# WORDS THAT MAKE LANGUAGE MODELS PERCEIVE

## ABSTRACT

Large language models (LLMs) trained purely on text ostensibly lack any direct perceptual experience, yet their internal representations are implicitly shaped by multimodal regularities encoded in language. We test the hypothesis that explicit sensory prompting can surface this latent structure, bringing a text-only LLM into closer representational alignment with specialist vision and audio encoders. When a *sensory prompt* tells the model to 'see' or 'hear', it cues the model to resolve its next-token predictions as if they were conditioned on latent visual or auditory evidence that is never actually supplied. Our findings reveal that lightweight prompt engineering can reliably activate modality-appropriate representations in purely text-trained LLMs.

## 1 INTRODUCTION

On its face, predicting the next word in web text appears orthogonal to perception. Language contains descriptions, but not the sensations themselves. Patel & Pavlick (2022) highlighted the difficulty of directly encoding the meaning of sensory inputs using language alone. This tension echoes the *symbol-grounding problem*, which asks how purely textual symbols can acquire intrinsic meaning without being anchored in direct perceptual experience (Harnad, 1990).

For LLMs, this becomes a question of whether they are merely manipulating surface statistics of text or encoding knowledge that connects text to the sensory world (i.e., are LLMs grounded?). One way to test this is by measuring how closely their embeddings align with those of models trained explicitly on sensory data. By defining the meaning of a symbol through the relationships it maintains with others (Wittgenstein, 1953), alignment can be quantified through *kernel-based representational similarity metrics* (e.g., mutual $k$-nearest neighbors). In this view, if the geometry of an LLM's representations resembles that of a vision model, then it encodes text in a way that is closer to the visually grounded representation. Huh et al. (2024) demonstrated that as models become more capable in their respective modalities, their kernel structures become more similar. They argue that this convergence reflects the existence of a shared latent structure underlying different modalities.

While such cross-modal convergence emerges with scale, it raises an interesting question. Instead of treating alignment as a fixed property of a model, can we elicit it at inference time? And if so, can even text-only models be controllably steered into perceptually grounded representations? Our results suggest that the answer to both is yes. We find that:

> A simple cue like asking the model to 'see' or 'hear' can push a purely text-trained language model towards the representations of purely image-trained or purely-audio trained encoders.

A model's representation can be understood as the embeddings it assigns to a set of inputs. Typically, these are taken from single forward passes. In this work, we introduce the notion of *generative representations*: when an LLM is asked to generate, each output token involves another forward pass, which recursively builds a representation that is not only a function of the prompt, but also of the sequence generated so far. We observe that these autoregressive steps yield a kernel representation that is more similar in geometry to vision and audio encoders. Moreover, we can control this generative representation; with an added sensory prompt, the resulting representation yields even higher alignment. To explain this intuitive, yet unexpected effect (Figure 1), we posit that an LLM implicitly maintains uncertainty over the kinds of evidence—visual, auditory, or otherwise—that

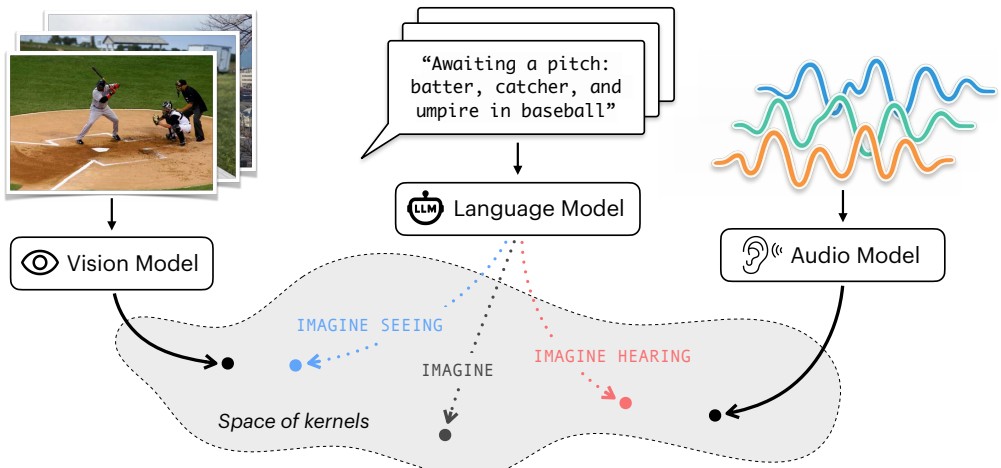

Figure 1: A cue that asks the model to 'see' (or 'hear') the provided text description moves the kernel representation of the model closer to the specialist model given the image (or audio) modality.

could have produced the text it is reading. When the context begins with an explicit cue such as 'see' or 'hear', the model conditions its generations on a specific sensory interpretation of the context. This means that representations are not fixed by training, but can be refined as a model reasons. In other words, there are ways to extract more perceptually grounded representations from LLMs trained only on text; we just need to know how to elicit it.

We quantify how sensory prompting steers the representation of an LLM by comparing them to frozen unimodal encoders in vision and audio domains. In our results, we find that:

- A single sensory word in the prompt can, through generation, shift the kernel of a text-only LLM closer to the geometry of sensory encoders.

- Representational similarity increases with generation length, as longer continuations give the model more opportunity to elaborate modality-specific content.

- Larger models exhibit higher alignment under sensory prompting and stronger modality separation.

- Visual cues allow LLMs to perform better on text-based visual reasoning questions.

## 2 METHODS

We evaluate how sensory prompts change the geometry of representations produced by text-only LLMs to resemble those of unimodal vision and audio encoders. To capture what the model represents as it generates, we incorporate generation into the representation. We then compare text- and sensory-induced kernels on paired datasets to quantify alignment, and we extend the analysis across additional models and datasets (Appendix C).

### 2.1 EXTRACTING GENERATIVE REPRESENTATIONS

We average hidden states from autoregressive continuations rather than using a single-pass embedding. Given a prompt $p$ and caption $c$, let $x_{1:T_0} = [p\|c]$ denote the input prefix with $T_0$ tokens. At generation step $t \geq 1$, the model has seen

$$x_{1:T_0+t} = [p\|c\|y_{1:t}], \qquad h_{T_0+t}^{(\ell)} = f_{\text{text}}^{(\ell)}(x_{1:T_0+t}), \qquad y_{t+1} \sim \text{Decode}\left(h_{T_0+t}^{(L)}\right),$$

where $y_t$ is the $t$-th generated token, $h_{T_0+t}^{(\ell)}$ is the hidden state of the final token at layer $\ell$, and $L$ is the total number of layers.

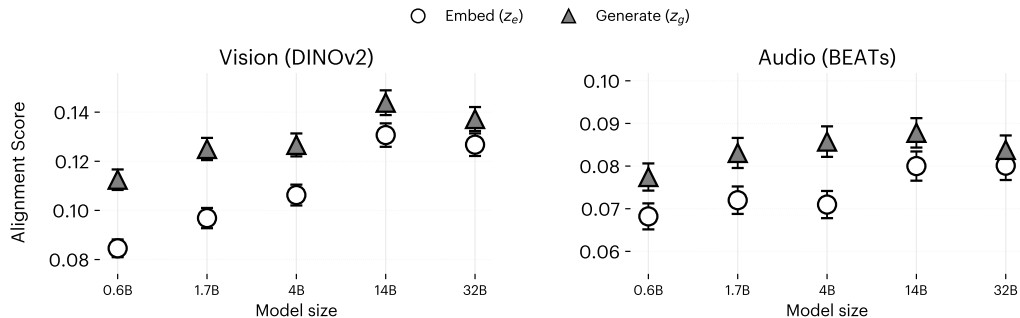

Figure 2: Generative representations (no cue from Figure 3b, 128 tokens) yield higher alignment than single-pass embeddings in language models. Left: alignment with vision encoder. Right: alignment with audio encoder.

We define two caption-level representations:

$$z_e = \frac{1}{LT_0} \sum_{\ell=1}^{L} \sum_{i=1}^{T_0} h_i^{(\ell)}, \qquad z_g(T) = \frac{1}{LT} \sum_{\ell=1}^{L} \sum_{t=1}^{T} h_{T_0+t}^{(\ell)}.$$

Here $z_e$ is a "single-pass" embedding that averages hidden states over all layers and all tokens in the initial prefix $[p\|c]$, while $z_g$ averages over all layers and the $T$ generated tokens only. Both $z_e$ and $z_g$ therefore map each caption $c$ to a single vector in the LLM representation space.

Residual connections in the LLM architecture make these averages a meaningful summary of the model's overall state, which we evaluate in Appendix A.

## 2.2 QUANTIFYING REPRESENTATIONAL SIMILARITY

For each image–caption (or audio–caption) pair, we compute an LLM embedding (either $z_e$ or $z_g(T)$) and a sensory embedding by taking the final-layer features from the vision or audio encoder and averaging across spatial or temporal tokens. Following the Platonic Representation Hypothesis framework (Huh et al., 2024), we define a *representation* as the set of embeddings a model produces on a dataset, and its induced *kernel* as the similarity structure among these embeddings. Given embeddings $\{z_i\}_{i=1}^n$, we define a kernel by $K_{ij} = \cos(z_i, z_j)$ and define $N_k^K(i)$ as the top-$k$ neighbors of $i$ under $K$. To compare two kernels $K, K'$, we use mutual-$k$NN alignment,

$$\text{Align}(K, K') = \frac{1}{n} \sum_{i=1}^{n} \frac{|N_k^K(i) \cap N_k^{K'}(i)|}{k},$$

where higher scores indicate higher representational similarity, i.e., two models are more aligned. For each prompt condition and dataset, we embed all samples, construct kernels from cosine neighbors, and compute alignment between the LLM and the corresponding sensory encoder. Error bars in paper figures denote $\pm 1$ bootstrap standard error ($B = 1000$), obtained by resampling $N$ paired rows with replacement from the dataset to form bootstrap replicates and recomputing the mutual-$k$NN alignment score. This captures the variability of the score under resampling of the data.

## 2.3 MODELS

All models are kept frozen during evaluation.

**Sensory Encoders:** For vision, we use DINOv2-Base (ViT-B/14, 768-dim) (Oquab et al., 2023), a self-supervised model trained only on images. For audio, we use BEATs-Iter3 (Chen et al., 2022), a self-supervised model trained only on natural sounds (AudioSet).

**Language Models:** We evaluate frozen Qwen3 LLMs (Yang et al., 2025) across scales (0.6B, 1.7B, 4B, 8B, 14B, 32B). These models are trained only on text, with no vision or audio supervision.

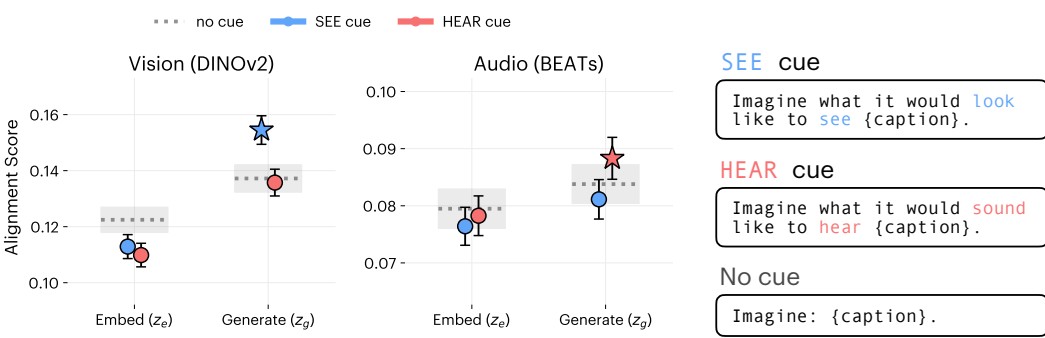

(a) Sensory cued alignment using single-pass embedding ($z_e$) and generative ($z_g$) representations.

(b) Prompt templates used in generative representations.

Figure 3: Sensory cues induce a generative text-only LLM representation that has higher alignment with the corresponding encoder. The star denotes matching cue-modality in generative representation.

## 2.4 DATASETS

We evaluate on image–caption and audio–caption datasets. **WiT** (Srinivasan et al., 2021): 1024 image–caption pairs from Wikipedia (as in Huh et al. (2024)). **AudioCaps2.0** (Kim et al., 2019): 975 audio–caption pairs from AudioSet.

## 3 RESULTS

### 3.1 GENERATIVE REPRESENTATIONS YIELD HIGHER ALIGNMENT

We first compare alignment based on single-pass embeddings with alignment from generative representations, where the LLM continues each caption for 128 tokens under the no-cue template (Figure 3b). As shown in Figure 2, simply allowing the model to elaborate on the caption already produces embeddings that align more closely with sensory encoders.

This suggests that prior work such as Huh et al. (2024), which evaluated alignment only from single-pass embeddings, underestimate how much cross-modal similarity is present in LLMs. Generation creates a representation that yields higher alignment—even without explicit sensory cues. It offers a way to achieve such alignment at inference time, without requiring additional training.

### 3.2 SENSORY CUES STEER GENERATIVE REPRESENTATIONS

We find that explicit sensory cues in the prompt can steer generative representations ($z_g$) to achieve higher alignment scores than under naive generation (i.e., when prompted without a sensory cue). We evaluate Qwen3-32B on paired image–text (WiT) and audio–text (AudioCaps) datasets by prepending each ground-truth caption with either a no cue baseline, a SEE cue, or a HEAR cue (Figure 3b). Each prompt generates a 128-token continuation, snippets of which are shown in Figure 4.

Figure 3a shows that the SEE cue increases alignment with the vision encoder (DINOv2) and decreases alignment with the audio encoder (BEATs). Conversely, the HEAR cue increases alignment with BEATs while reducing alignment with DINOv2. These results indicate that a single sensory cue in the prompt can steer the internal representations of the LLM to better match the geometry of the modality the cue invokes. In contrast, we find that sensory prompting cannot steer single-pass embedding representations ($z_e$) in the same way. Inserting sensory cues into the prompt decreases the alignment from the no cue prompt. Thus, the higher alignment achieved through sensory prompting is a result of the representation formed during generation. We validate this result on additional sensory encoders, language models, and datasets in Appendix C.

Mutual-$k$NN provides an interpretable illustration of when sensory prompting helps shift the LLM's representation toward the intended modality: Figure 5 shows two examples from WiT where the SEE cue yields the largest increase in shared top-$k$ neighbors with the vision encoder. We consider

| Image / `Caption` | SEE cue generation | HEAR cue generation |
|---|---|---|
| `Awaiting a pitch; batter; catcher, and umpire in baseball` | ...I need to visualize a baseball field. The batter stands at the home plate, right? They're holding the bat, probably in a ready position, feet shoulder-width apart. Their body language should show they're focused, maybe squinting at the catcher or the umpire... | ...There's the general ambient noise of the stadium: crowd murmurs, maybe some distant cheers or chatter. The crack of a bat from another field if it's a doubleheader or a minor league game nearby. The sound of players moving, like cleats scraping on the dirt, gloves being adjusted... |
| `Detail of the stained glass windows at the Chapel of St Saviour` | ...Let me start by recalling what I know about stained glass in chapels. Usually, they have intricate designs with biblical scenes, saints, maybe some geometric patterns. The colors are vibrant because of the glass, and when light passes through, it creates a colorful effect... | ...Hmm, stained glass windows are usually visual, so translating that into sound is an interesting challenge ... glass, lead, maybe some metal reinforcements. Each material has its own acoustic properties. Glass can make a ringing sound when struck, like a bell... |
| `Groundnut kernel` | ...I should think about the color. When you open a peanut shell, the kernel is usually a pale yellow or light brown, maybe with a slight reddish tinge. But I'm not entirely sure if there are different varieties with different colors... | ...Let me start by breaking down the process. When you crack a peanut shell, there's a distinct sound—crunchy, maybe a bit sharp. But the kernel itself, once the shell is off, is softer. If you bite into it, it's more of a soft crunch, maybe with some give... |

Figure 4: Snippets of text generated from WiT captions, under sensory cues. *We highlight, by hand, words that may be associated with the sensory modality.* Full example found in Appendix G.1.

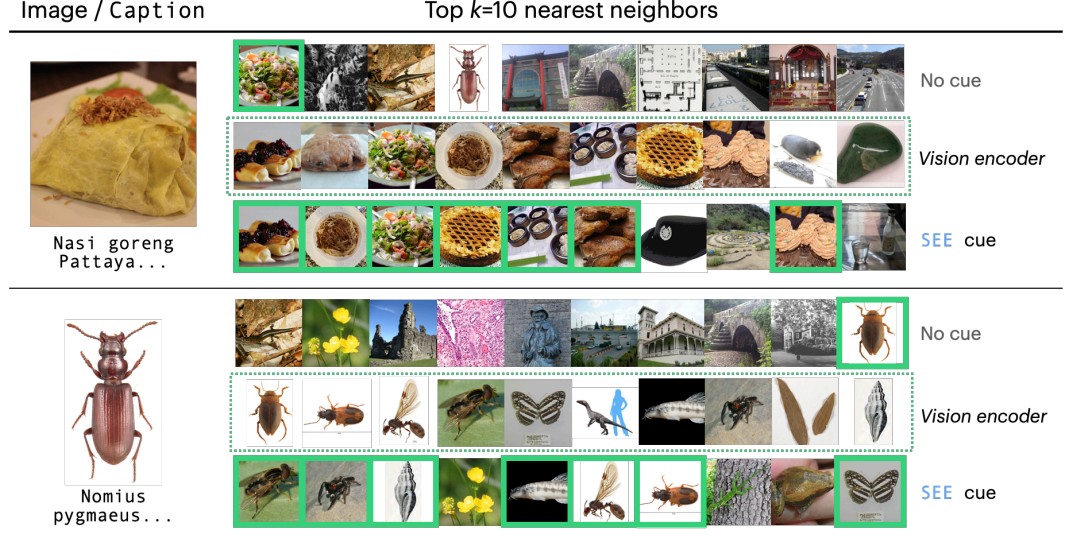

Figure 5: Selected examples where visual prompting yields the largest increase in shared top-$k = 10$ neighbors with the vision encoder (vs. no cue). Green outlines mark inputs also among the vision encoder's nearest neighbors. Additional generations and examples appear in Appendix B.

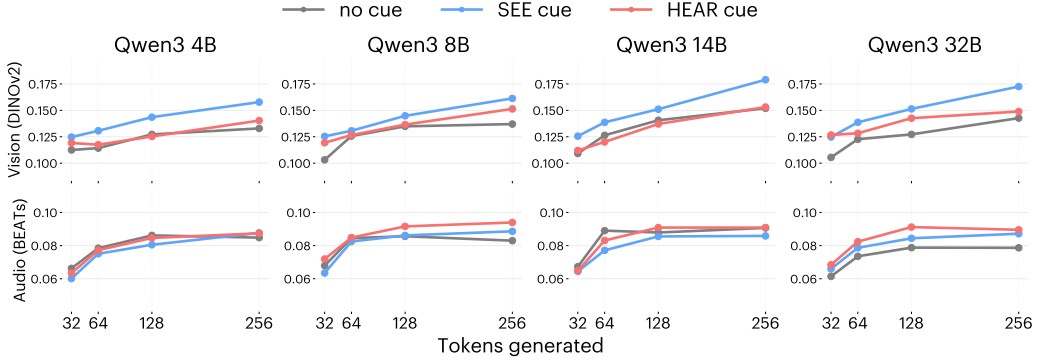

Figure 6: Alignment to sensory encoders increases with generation length.

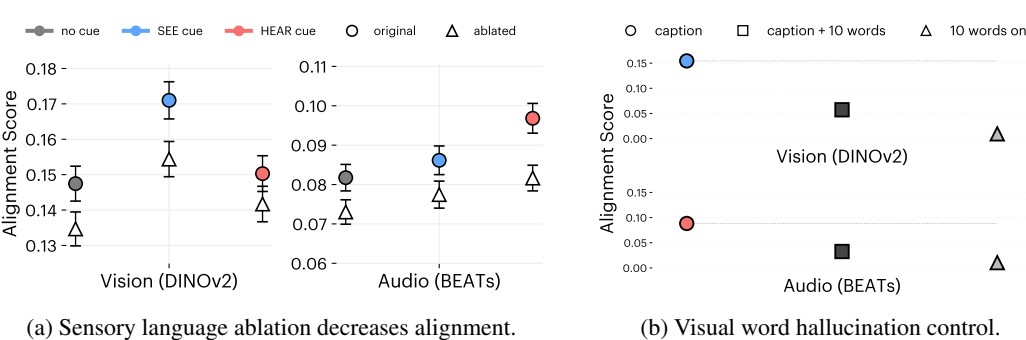

(a) Sensory language ablation decreases alignment.

(b) Visual word hallucination control.

Sensory ablation

```
Please rewrite the following text by removing all sensory-specific words or descriptions
(e.g., related to sound, sight, smell, touch, taste) and replacing them with neutral, non-
sensory words; preserve the event or action while removing explicit sensory grounding: {text}.
```

(c) Prompt template

Figure 7: Correct sensory language is necessary for increase in alignment. Error bars not visible in (b) due to scale.

the caption "Nasi goreng Pattaya." Under the no cue condition, the generation describes general information (e.g., *"Nasi goreng Pattaya is a local delicacy from Pattaya, Thailand, but it's also popular in neighboring countries"*), and the nearest neighbors include "Yam thale (Thai dish)", "Lankascincus gansi (a skink species in Sri Lanka)", and the "Korean–Chinese Cultural Center in Incheon, South Korea". These neighbors plausibly arise from the model's emphasis on geographic and cultural descriptors in the no cue text, which happen to be less visually related. By contrast, the SEE cue shifts the continuation toward concrete food description (*"the main components: fried rice, omelette, and the sauce... toppings like shrimp, chicken, or vegetables"*). Its nearest neighbors under this condition are themselves food items—such as "Blinchiki filled with cheese and topped with blackberries" and "Spaghetti topped with pulled pork in a marinara sauce..."—showing that by emphasizing visual descriptors of the food, the LLM produces a representation that aligns more closely with the vision encoder's representation of the corresponding image. Additional qualitative examples (including those that decrease the overlap in nearest neighbors) can be found in Appendix B.

## 3.3 EDITING SENSORY-CUED GENERATIVE REPRESENTATIONS

Having shown that sensory cues condition generation such that its representation is more reflective of a given modality, we now explore which aspects of the generations are responsible for this effect. Generative hidden states are defined over the model's own outputs, and thus, editing the generation amounts to editing the representation itself. This allows us to test how alignment depends on specific language choices. We find that a *lack of sensory words* lowers alignment, but not just any sensory words increase it—what matters is using *scene-appropriate sensory details*.

**Sensory-word ablation.** To determine whether alignment depends on the explicit use of modality-specific language, we perform a sensory-word ablation on 256-token generations from Qwen3-32B using the prompts in Figure 7c. We choose longer generations to ensure that the original outputs contain sufficient sensory references for a meaningful intervention. Importantly, the ablation preserves the semantic content of each generation while replacing modality-specific language with neutral phrasing (see Appendix G.3 for examples). Following ablation, alignment to both vision and audio encoders drops significantly (Figure 7a), thus, sensory language is necessary for the observed alignment.

**Controlling for hallucinations.** However, sensory language itself is not sufficient. Mutual-$k$NN captures relational similarity: it evaluates whether a caption and an image (or audio) induce similar neighborhoods over a dataset. To show that observed alignment gains do not arise from "hallucinations," where generic modality-specific words are added rather than attributes that accurately describe

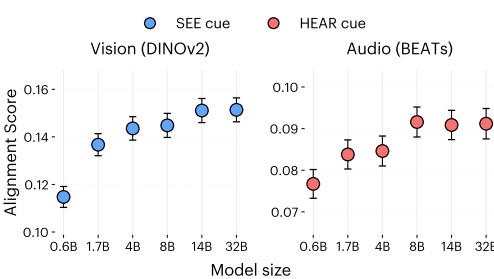 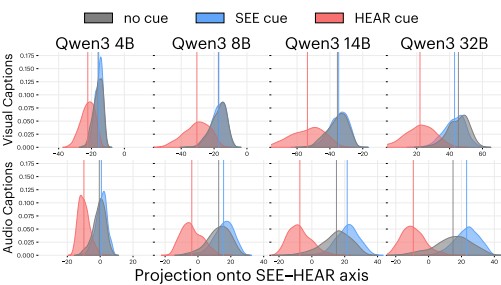

(a) Larger models are more aligned to sensory encoders under corresponding cues.

(b) Embedding projections onto visual–auditory axis show clearer modality separation in larger models.

Figure 8: Stronger sensory alignments and modality separations emerge in larger models.

the given sample, we edit captions with additional visual words. That is, the mere presence of such visual descriptors could not increase alignment to a vision encoder. We sampled 10 random visual attributes from the 45,092 object properties parsed from Visual Genome (Krishna et al., 2017), and constructed variants of each caption from WiT in the form `{caption} → {caption + 10 random visual words}` and `{caption} → {10 random visual words}`. In Figure 7b, we find that alignment decreases when captions are appended with random visual words, and drops further when captions are replaced entirely by them. This indicates that the observed gains do not simply arise from hallucinations of modality-specific vocabulary, but instead reflect that mutual-$k$NN captures relational structure tied to scene-appropriate sensory detail. That is, sensory cues steer LLMs toward a correct modality-specific generation that brings caption–caption relations in the LLM closer to those of vision or audio models.

### 3.4 GENERATION LENGTH IMPROVES SENSORY ALIGNMENT

In Figure 6, we find that alignment to vision and audio encoders increases as the LLM generates more tokens, suggesting that longer outputs allow the model to elaborate modality-specific content. In constrast, at shorter lengths (32 and 64 tokens), the LLM often produces little beyond restating the prompt (e.g., 'Okay, the user wants me to imagine...').

Interestingly, even mismatched cues (e.g., SEE prompts evaluated on audio alignment) can outperform the no cue baseline. For instance, at 256 tokens, Qwen3-32B achieves better alignment to both modalities under either cue than under a neutral prompt. We interpret this as an effect of shared cross-modal structure: many sounds (e.g., snoring, barking) have associated visual features, so visually descriptive generations can still increase alignment with auditory encoders. However, we note that alignment can decline as you continue to increase output tokens due to semantic drift from the prompt (Appendix A).

### 3.5 SENSORY ALIGNMENT SCALES WITH LARGER MODELS

We find that larger models have higher alignment with vision and audio encoders under appropriate sensory cues (Figure 8a; 128-token generations). Moreover, cue-specific representations become more separable with scale (Figure 8b). To quantify this, we project embeddings $\mathbf{x}_i \in \mathbb{R}^d$ onto a sensory axis defined by the mean difference between prompt conditions. Let $\mu_{\text{SEE}} = \frac{1}{N}\sum_i x_i^{\text{SEE}}$, $\mu_{\text{HEAR}} = \frac{1}{N}\sum_i x_i^{\text{HEAR}}$, be the mean embeddings under each cue. We define $\mathbf{v} = \frac{\mu_{\text{SEE}} - \mu_{\text{HEAR}}}{\|\mu_{\text{SEE}} - \mu_{\text{HEAR}}\|}$, and compute projections $s_i = \mathbf{x}_i^\top \mathbf{v}$, giving a scalar position along the visual–auditory axis. We estimate the distribution of $s_i$ using kernel density estimation.

This result reflects more separate modality-specific representations with increasing scale (see Figure 31 for the same evaluation on DCI). In smaller models, no cue generated embeddings consistently resemble those generated by SEE, even when the caption describes a sound, suggesting that language models tend to default to a visual framing without an explicit cue. As model size increases, however, no cue embeddings shift closer to those generated by HEAR.

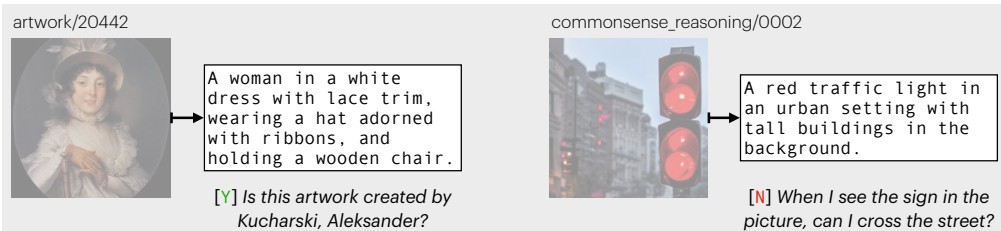

Figure 9: Instead of answering from an (`image`, `Q`) pair as in standard VQA, the model receives a (`caption`, `Q`) pair, where the caption is a text projection of the image.

Table 1: Visual prompting applied to the MME benchmark projected to text.

| | artwork | celebrity | code_reasoning | color | commonsense | count | existence | landmark | numerical | position | posters | scene | text_translation | Overall |
|---|---|---|---|---|---|---|---|---|---|---|---|---|---|---|
| No cue | 59.92 | 50.49 | 92.50 | 81.67 | **75.48** | 70.56 | 83.33 | 52.42 | 70.00 | 49.45 | 72.00 | 70.33 | 90.83 | 64.08 ± 1.03 |
| SEE | **60.83** | **50.98** | **94.17** | **83.33** | 75.48 | 73.89 | **86.67** | **52.83** | **72.50** | **52.78** | 75.62 | **72.83** | **93.33** | **65.74 ± 1.12** |
| $n$ | 400 | 340 | 40 | 60 | 140 | 60 | 60 | 400 | 40 | 60 | 294 | 400 | 40 | 2334 |

### 3.6 VISUAL QUESTION ANSWERING IN TEXT SPACE

We test whether sensory prompting meaningfully creates better visual representations in terms of downstream task performance. To answer this, we test whether sensory cues allow language models to perform more accurate visual reasoning in the text modality, we adopt the "VQA without V" setting from Chan et al. (2025); Chai et al. (2024). Instead of providing an (image, question) pair as in standard VQA, we provide a (caption, question) pair, where captions serve as projections of images into text space. The model is then tasked with answering yes/no questions based solely on these captions. We use the MME benchmark (Fu et al., 2023a) and first caption all images with Qwen2.5-VL-3B-Instruct (Qwen et al., 2025). Because OCR questions require recognizing text directly from the image rather than reasoning over the caption, we exclude the OCR category from our evaluation. We then evaluate the question-answering language model under two prompt conditions: a neutral instruction and a visual framing, which explicitly asks the model to *imagine seeing* the caption before answering. For scoring, we extract only the categorical yes/no answer from the model's output. The full prompt can be found in Appendix F.

Averaged over three Qwen-family language models (Qwen3-14B, Qwen2.5-Instruct-7B, Qwen2.5-Instruct-14B), the no-cue condition achieves 64.08 ± 1.03% accuracy, while the SEE condition achieves 65.74 ± 1.12% accuracy (mean ± SE across models), corresponding to an average gain of 1.7 ± 0.4% (Table 1). These findings support the view that language models can act as text-space vision models, and that simple sensory cues improve performance specifically where reasoning about the imagery helps disambiguate the caption.

## 4 RELATED WORK

Alignment between LLMs and models trained on modalities grounded in sensory data has been observed several times in past work, even though LLMs only experience the world through text. Abdou et al. (2021) demonstrate that the geometry of color word embeddings in LLMs aligns with human perception of these colors. Patel & Pavlick (2022) show that text-only LLMs can generalize structured concepts from the physical world, such as spatial directions, allowing them to reason about navigation with terms like "left" and "right" despite never having direct perceptual experience. These results are consistent with the observation that models trained on different modalities converge in their representation as the models scale (Huh et al., 2024).

Notably, some LLM representations fail to capture sensory structure. Xu et al. (2025) find that text-only models encode abstract properties of words but perform poorly on sensory and motor features when rating words. However, Pavlick (2023) argue that the lack of direct grounding does not

mean LLMs are unable to represent meaning. That is to say, weak sensory representations do not rule out the possibility of better ones. Our results provide such a case: sensory cues steer generative representations toward the target modality even when single-pass embeddings do not.

Gu et al. (2023) show that models can learn to solve visual tasks using only language supervision, demonstrating that captions can act as proxies for images. This connects directly to our VQA experiment, where we test whether sensory cues improve text-only models' ability to reason about captions as if they were images. Ashutosh et al. (2025) further show that through iterative feedback from a vision/audio model, a text-only LLM can perform multimodal captioning and generation.

## 5 DISCUSSION

Perception involves both the reception of stimuli through sensors and their interpretation in context. We have shown that cueing a text-only language model to imagine a specific modality shifts its internal representation toward that of an explicit sensory encoder, which has a representation reflective of the true sensory structure because it is trained on that modality. When prompted to "see" (or "hear") the model behaves as if its input were grounded in perceptual evidence—producing modality-appropriate responses shaped by the generation of accurate visual (or auditory) imagery. This is quantified by comparing the kernel induced over captions by the LLM to the kernel induced by a sensory encoder over paired data; higher alignment under sensory prompting means that caption–caption relationships are more like those in the vision (or audio) model.

Our work extends prior observations of passive cross-modal convergence by showing that alignment can also be actively steered at inference time. This supports a view of language models as implicitly multimodal agents: their representations encode a distribution over possible latent causes, including sensory ones, for the text they process. Importantly, generative representations allow us to induce a kernel over a set of text inputs according to a prior specified in the prompt. In our case the prior is sensory, but in principle it could be other characteristics—for example, spatial layout or sentiment. This means prompting allows an interpretable way to steer which relationships the kernel encodes, rather than leaving them implicit to the model.

Sensory prompting offers a practical implication: since the cues are human-specified and generations are semantically meaningful, they provide an interpretable way for extracting modality-steered embeddings from text-only models. These embeddings could support tasks such as cross-modal retrieval or distillation. More broadly, it suggests that the line between unimodal and multimodal models is less rigid than often assumed. Sensory prompting also fits alongside chain-of-thought and retrieval cues as part of a growing toolkit for inference-time control, showing that what a model represents is not fixed at training but can be elicited through context.

## 6 LIMITATIONS

While we have determined that text-based sensory cues can increase representational alignment to vision and audio encoders at a scale comparable to Huh et al. (2024), we have not fully explored the degree to which this alignment can be improved. In particular, we focus primarily on lightweight cues such as 'see' and 'hear', but do not explore broader variation in instruction phrasing. We evaluate sensory prompting with null prompts and with other verbs in Appendix A.

Furthermore, we note that LLM alignment to audio encoders is lower than alignment to vision encoders and less reliably steerable. One explanation is that audio encoders like BEATs learn low-level acoustic patterns (e.g., frequency, rhythm, timbre) that map less directly to language, whereas vision encoders such as DINOv2 capture object- and scene-level features that align more naturally with words. Supporting this, BEATs variants fine-tuned with AudioSet labels achieve much higher alignment (Appendix C). We also find that steerability depends on the dataset: alignment is more reliably steerable on shorter, under-specified captions such as WIT and COCO, and smaller on visually detailed captions such as DCI (Appendix C). Finally, sensory prompts may encourage the model to hallucinate specific perceptual details in the generation that are not actually supported by the input. While this is acceptable in generative contexts, it may be problematic in settings requiring factual visual or auditory precision.

## THE USE OF LARGE LANGUAGE MODELS

We used large language models (LLMs) as assistive tools during the writing of this paper. We use LLMs to improve the conciseness of writing, as well as to help generate code for making figures. All conceptual contributions, experimental design, and interpretation of results were carried out solely by the authors. The authors take full responsibility for the content of this paper.

## ETHICS

This work uses only publicly available datasets and open-source pretrained models. We do not collect any human subject data. All datasets have usage licenses that permit academic research, and our use is consistent with their intended purposes. No sensitive information is involved.

## REPRODUCIBILITY

All experiments in this paper are based on publicly available datasets and open-source pretrained models. Our implementation of the alignment metric follows the procedure introduced in Huh et al. (2024), whose code is openly released. We will provide our full codebase upon request and make it publicly available at the time of release.

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

APPENDIX CONTENTS

## A    EXTENDED ANALYSIS OF SENSORY PROMPTING

**Additional instruction verbs.**    To assess the robustness of sensory prompting, we replicate the experiment using a broader set of $n = 10$ instruction verbs: *"conceptualize", "consider", "describe", "detail", "explain", "formulate", "imagine", "think (about)", "wonder", "write"*. We also include a null baseline where instead of an instructional sensory prompt, we prepend a random sentence drawn from captions of the DCI dataset. Results averaged across verbs (mean $\pm$ standard error) are shown in Figure 10.

**Per-verb breakdown.**    In Figure 11 we provide alignment scores for each verb individually. Although overall trends are consistent, different verbs yield slightly different levels of alignment.

**Potential for prompt optimization.**    The space of possible instruction prompts is vast and we have only explored a subset. For example, *"describe"* allows for higher alignment than *"imagine"*, as shown in Figure 12. Also see Figure 13 for the trend across Qwen3 sizes and generation lengths. This highlights the potential for optimizing prompts to maximize alignment. A current limitation, however, is that alignment evaluation is computationally expensive, making systematic prompt search challenging.

**Instruction vs. non-instruction prompting.**    Finally, we compare instruction-style prompting (*"imagine what it would be like to see..."*) with analogous non-instructional forms (*"they imagined seeing..."*). As shown in Figure 14, alignment is much higher under the instruction form, indicating that instruction-tuned models are actively following the command rather than responding to the semantic content of the word itself.

**Overgeneration reduces alignment.**    While increasing generation length up to 256 tokens improves sensory alignment, we observe that performance can decline again at 512 tokens (Figure 15). This suggests that overly long generations may lead to semantic drift or off-topic elaboration. See Appendix G.4 for qualitative examples.

**Layer-wise evaluation.**    One concern is that the observed benefits of sensory prompting might reflect only *superficial priming* of the final layer: in other words, that the LLM is simply adjusting its last-step representation so the output head is biased toward modality-related words (e.g., "seeing" toward visual descriptors, "hearing" toward auditory ones), rather than inducing a deeper change in its internal representations. To test this, we compute alignment scores layer-by-layer rather than only on the mean-layer embedding.

Figure 16 shows layer-wise results for Qwen3 evaluated on both the WiT (image–text) and AudioCaps (audio–text) datasets. The sensory prompting trend is preserved across layers, indicating that the effect is not confined to superficial bias at the output layer but instead reflects a consistent shift in the model's intermediate representations. Interestingly, we also find that using the mean embedding across all layers often yields higher alignment than any single layer. One possible explanation is that averaging smooths out layer-specific noise while retaining complementary information across the hierarchy, though a full understanding of this phenomenon remains open for future study. Figure 16 shows layer-wise results for Qwen3, demonstrating that averaging captures consistent trends across layers.

**Redirecting sensory cues.** We generate 128-token outputs from Qwen3-32B using either a SEE or HEAR cue, then rewrite them with redirection templates (Figure 17b) that flip the modality. This produces a clear double dissociation: generations that redirect the cue from SEE to HEAR align more with audio encoders and less with vision, while generations that redirect the cue from HEAR to SEE shift toward vision encoders and away from audio (Figure 17a).

Note that redirection remains effective even when the initial output is lossy—for example, a visual caption prompted with HEAR may drop visual detail and introduce auditory description. Yet rewriting back to SEE restores alignment, suggesting that the model can correctly make cross-modal inferences (e.g., reconstructing visual structure from auditory framing). These results show that sensory prompts causally steer representations toward or away from modality-specific subspaces. Examples of redirected generations are provided in Appendix G.2.

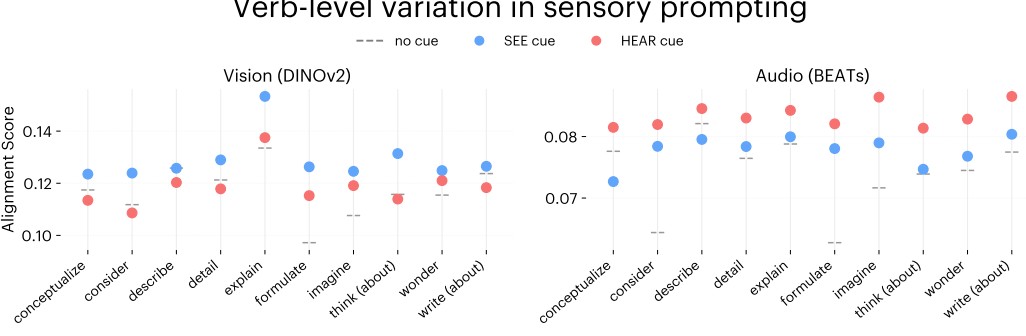

Figure 10: Extension of Figure 3 to additional verbs and null prompts (sentences drawn from DCI captions) for prompting.

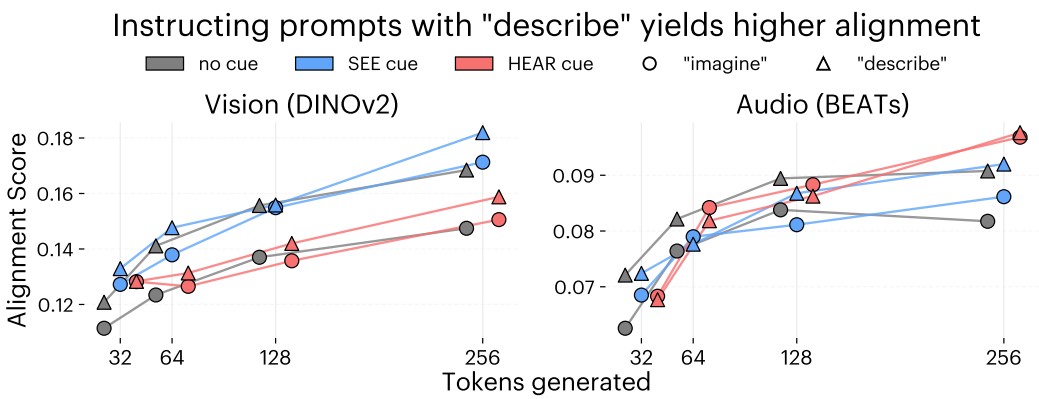

Figure 11: Extension of Figure 3 to additional verbs for prompting.

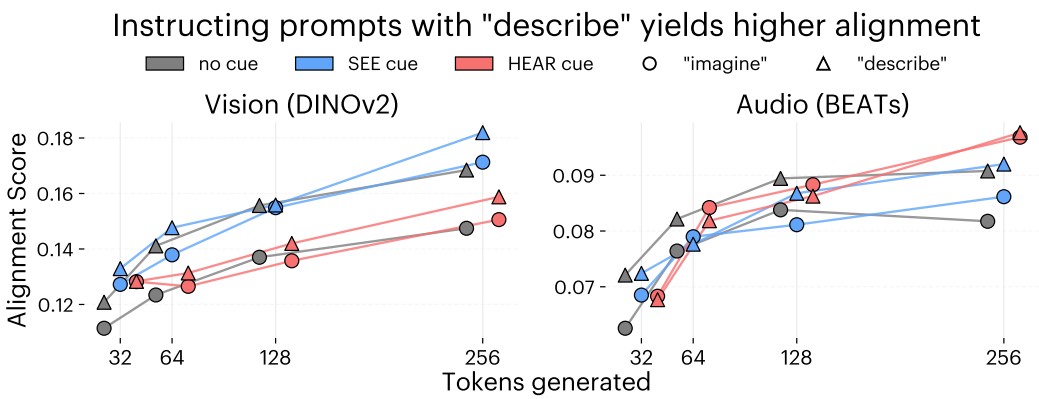

Figure 12: Instructing the LLM with "describe" can yield better alignment than "imagine".

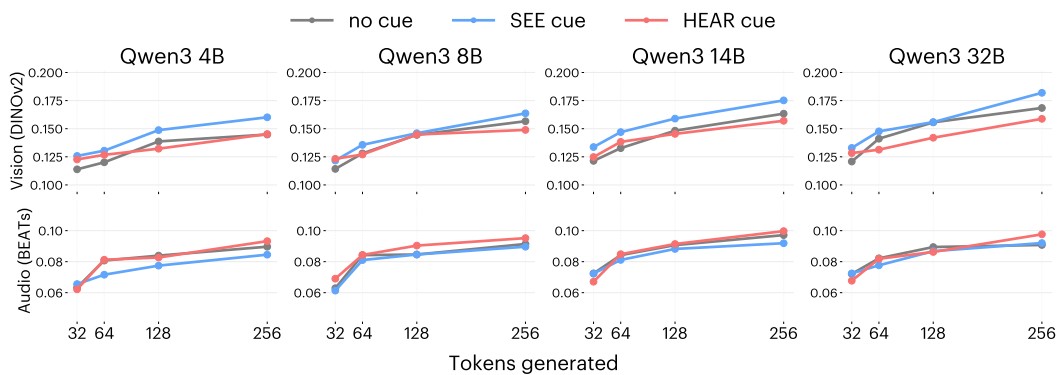

Figure 13: Extension of Figure 6 to "describe" instructed prompting.

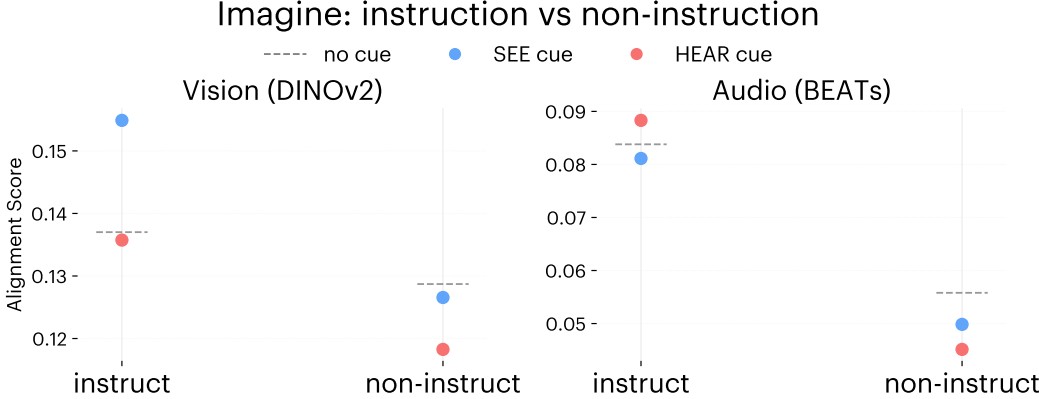

Figure 14: Alignment scores without instruction prompting.

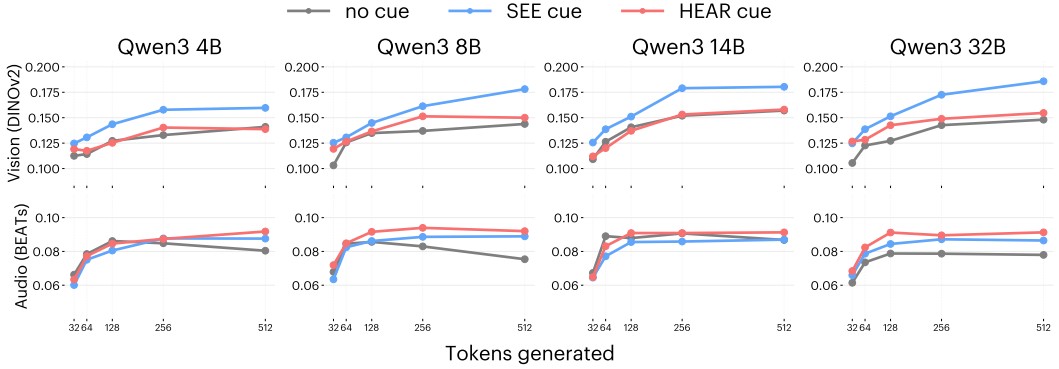

Figure 15: Extension of Figure 6 to 512-token generations.

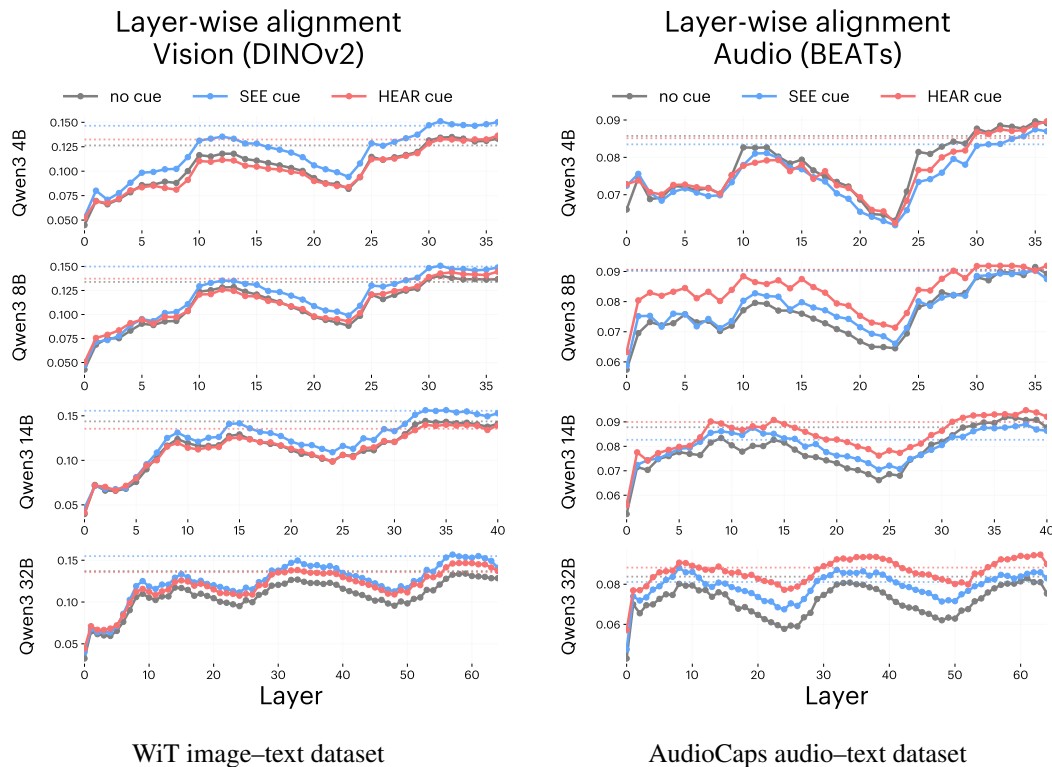

Figure 16: Layer-wise alignment with 128-token prompts. The sensory prompting effect is consistent across layers. The dotted line indicates alignment when using the mean embedding across all layers.

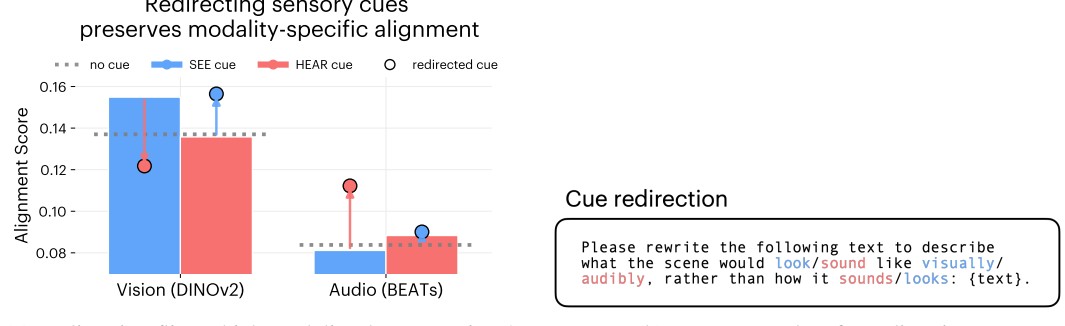

(a) Redirection flips which modality the generations' representations align with.

(b) Prompt template for redirection.

Figure 17: Redirecting sensory cues.

## B ADDITIONAL QUALITATIVE EXAMPLES

Mutual-$k$NN alignment also lets us examine where sensory prompting increases or decreases overlap with the sensory encoder.

Figure 18 shows four WiT captions where mutual-$k$NN overlap with the vision encoder shifts under sensory prompting. For "Nasi goreng Pattaya...", (Section B.1) the no cue output highlights cultural context (*"Nasi goreng Pattaya is a local delicacy from Pattaya, Thailand, but it's also popular in neighboring countries"*), retrieving neighbors such as "Yam thale" (Thai dish), "Lankascincus gansi" (a skink species), and the "Korean–Chinese Cultural Center." With the SEE cue, the continuation instead describes visual details (*"the main components: fried rice, omelette, and the sauce. . . toppings like shrimp, chicken, or vegetables"*), and the neighbors are foods such as "Blinchiki filled with cheese and topped with blackberries" and "Spaghetti topped with pulled pork in a marinara sauce with a barbecue sauce base," yielding higher overlap with the vision encoder.

A similar shift occurs for the stinking beetle caption "Nomius pygmaeus..." (Section B.2). The no cue continuation paraphrases the encyclopedic description, producing neighbors that mix insects with unrelated entries such as "Lankascincus gansi" (a skink species), "The buttercup (Ranunculus spp) occurs in many variations. . . ," and "Sawley Abbey, near to Sawley, Lancashire, Great Britain." With the SEE cue, the continuation instead emphasizes visible traits (*"a small, dark-colored beetle with a hard exoskeleton"*), and the neighbors become dominated by insect images such as "Anasimyia lineata" (hoverfly), "Pellenes seriatus" (spider), and "Winged caste of Huberia striata" (ant). This illustrates how visual prompting shifts the representation toward appearance-based similarity rather than encyclopedic associations.

In contrast, not all captions benefit from visual prompting. The bottom two rows show examples where visual prompting reduces alignment. The no cue continuation stays geographic, beginning with *"Okay, the user is asking about the 'Flag of ntice, Plze–South District.' First, I need to check if there's any existing information about a place called 'ntice' in the Plzeň–South District."* This framing retrieves sensible neighbors related to civic symbols such as "Nova tifta, Municipality of Sodraica, Slovenia," "Flag of Czech village Ln," and "Coat of arms of ievac." With the SEE cue, however, the continuation drifts into speculation about spelling and fictional names: *"Maybe they meant 'notice'? But that doesn't make sense . . . or perhaps 'Ntice' is a fictional or misspelled name."* As a result, the nearest neighbors shift away from flags toward mismatched entries such as "Planjava Northwest, seen from," "Coat of arms of ievac," and "Wine cellar in Szld." Here, visual prompting lowers overlap with the vision encoder by pulling the representation away from concrete geographic symbols and toward unrelated objects and places.

For the caption "MOLA map of Suess" (Section B.4), the no cue continuation stays on-topic, explaining that *"MOLA stands for Mars Orbiter Laser Altimeter . . . it created topographic maps of Mars by measuring the time it takes for a laser pulse to reflect off the surface and return to the spacecraft."* This technical framing retrieves neighbors tied to Mars imagery such as "This topographic map is created using Mars Orbiter Laser Altimeter (MOLA) technology . . . Cerulli crater," "Troughs and streaks in Arcadia quadrangle, as seen by hirise under HiWish program," and "King Lear Peak from Sulphur." By contrast, the SEE cue steers the continuation into speculative territory, with text like *"Suess is likely referring to the fictional land of Narnia . . . maybe the planet from The Hitchhiker's Guide to the Galaxy . . . or Dr. Seuss, the author."* The resulting neighbors ("Grotheer in 2018," "Planjava Northwest, seen from," "Daggett in 1984") lack clear visual connection to Martian maps. Here, visual prompting reduces overlap with the vision encoder by shifting the representation away from genuine topographic descriptions and toward unrelated associations.

Together, these examples highlight that sensory prompting increases alignment when it elicits modality-relevant descriptors, but can hurt when the cue introduces ambiguity or distracts from the grounded semantics of the caption.

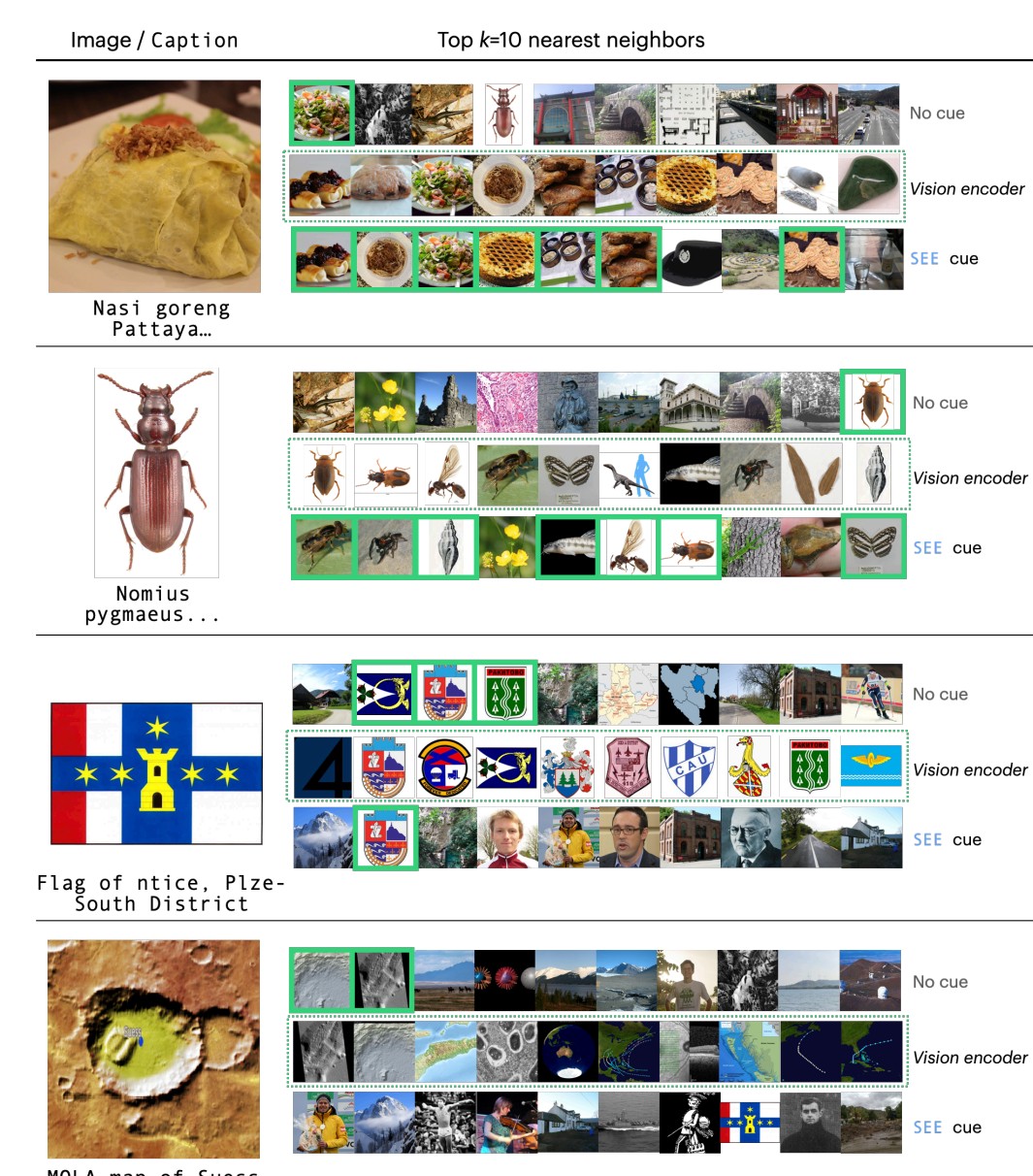

Figure 18: Extension of Figure 5 to additional examples.

### B.1 EXAMPLE: "NASI GORENG PATTAYA..."

> Caption:      Nasi goreng Pattaya (Pattaya fried rice), a
> local delicacy from Pattaya, Thailand.  It is fried
> rice served in a pouch of omelette, commonly found in
> Indonesia, Malaysia, Singapore and Thailand.

> **No cue output (128 token generation)**
>
> Okay, the user is asking about Nasi Goreng Pattaya, a dish that's fried rice served in an omelette pouch, commonly found in Indonesia, Malaysia, Singapore, and Thailand. They want the information in Bahasa Indonesia. Let me start by confirming the details.
>
> First, I should mention that Nasi Goreng Pattaya is a local delicacy from Pattaya, Thailand, but it's also popular in neighboring countries. The key points are the fried rice, the omelette pouch, and the regions where it's found. I need to make sure the translation is accurate and uses the correct terms in

> **SEE output (128 token generation)**
>
> Okay, the user wants me to imagine what Nasi Goreng Pattaya looks like, described in Bahasa Indonesia. Let me start by recalling what Nasi Goreng Pattaya is. It's a type of fried rice wrapped in an omelette pouch, popular in Southeast Asia. I need to make sure I get the ingredients and preparation right.
>
> First, the main components: fried rice, omelette, and the sauce. The fried rice is typically made with day-old rice, eggs, soy sauce, chili, and various toppings like shrimp, chicken, or vegetables. The omelette is made from beaten eggs

### B.2 EXAMPLE: "NOMIUS PYGMAEUS..."

> Caption: Nomius pygmaeus (Dejean).  This species is known
> under the vernacular name stinking beetle because of
> the strong fetid smell that the adults produce.  They
> are attracted to lights and sometimes find their way
> into houses.  It was reported in the literature that
> at one occasion an entire village had to be evacuated
> because of the odor produced by these small beetles.
> The species was often listed as very common at light
> in the xix Century but is rare today.  The species
> has an unusual range being found in North America
> and Europe and there is no evidence that it was
> transported by man from one continent to the other.

> **No cue output (128 token generation)**
>
> Okay, so I need to figure out what the user is asking about Nomius pygmaeus. Let me start by reading through the provided information again to make sure I understand the context. The beetle is called the stinking beetle because of its strong odor, and it's attracted to lights. There's a story about a village evacuation due to the smell, and it was common in the 19th century but rare now. Also, it's found in both North America and Europe without evidence of human transport.
>
> Hmm, the user might be asking about the reasons behind its rarity now, or maybe the history of its

> **SEE output (128 token generation)**
>
> Okay, so I need to imagine what it would look like to see Nomius pygmaeus, the stinking beetle. Let me start by recalling what I know about beetles in general. Beetles are insects with hard shells, so I can picture a small, maybe dark-colored beetle. The user mentioned that it's called the stinking beetle because of its strong odor. That must be a key feature.
>
> First, the appearance: The user didn't give specific details, but since it's a beetle, I can assume it has an exoskeleton, maybe some coloration. The name "pygma

### B.3 EXAMPLE: "FLAG OF NTICE, PLZE-SOUTH DISTRICT"

Caption: `Flag of ntice, Plze-South District`

**No cue output (128 token generation)**

Okay, the user is asking about the "Flag of ntice, Plze-South District." First, I need to check if there's any existing information about a place called "ntice" in the Plzeň-South District. Plzeň-South is a district in the Czech Republic, part of the Plzeň Region. I know that Plzeň is a major city there, but I don't recall any municipality named "ntice." Maybe it's a misspelling or a very small village?

Let me think. Sometimes, place names can be misspelled or have alternative spellings.

**SEE output (128 token generation)**

Okay, the user is asking me to imagine what the flag of the "ntice, Plze-South District" would look like. First, I need to figure out if there's a typo here. "ntice" doesn't seem right. Maybe they meant "notice"? But that doesn't make sense as a place name. Alternatively, could it be "Ntice" as a fictional or misspelled name? Or perhaps "Notice" is part of the district's name? Let me think.

Wait, Plze-South District might be a reference to the Plzeň region in the Czech Republic.

### B.4 EXAMPLE: "MOLA MAP OF SUESS"

Caption: `MOLA map of Suess`

**No cue output (128 token generation)**

Okay, so I need to imagine a MOLA map of Suess. First, I should figure out what MOLA means. I remember that MOLA stands for Mars Orbiter Laser Altimeter, which was an instrument on the Mars Global Surveyor. It created topographic maps of Mars by measuring the time it takes for a laser pulse to reflect off the surface and return to the spacecraft. So a MOLA map would be a topographic map showing elevation data, with different colors representing different heights.

Now, Suess... Wait, Suess isn't a place on Mars that I'm familiar with. Maybe it's

**SEE output (128 token generation)**

Okay, the user is asking to imagine what a MOLA map of Suess would look like. First, I need to clarify what they mean by "Suess." Suess is likely referring to the fictional land of Narnia from C.S. Lewis's books, but wait, no, that's Narnia. Maybe they meant the planet from the book "The Hitchhiker's Guide to the Galaxy," but that's Magrathea. Wait, maybe it's a typo. Alternatively, "Suess" could be a misspelling of "Suess" as in Dr. Seuss, the author

## C  EVALUATION ON ADDITIONAL MODELS AND DATASETS

We include additional alignment results in Figures 19–24 using a broader set of sensory encoders extended to additional datasets. For vision, we evaluate DINOv2 (Oquab et al., 2023), CLIP (Radford et al., 2021), ViT-MAE (He et al., 2022), and ViT-MSN (Assran et al., 2023). For audio, we evaluate CLAP (Elizalde et al., 2023), BEATs (Chen et al., 2022), BEATs+ (BEATs fine-tuned on AudioSet labels), Audio-MAE (Huang et al., 2022), and EAT (Gong et al., 2023). We also extend to additional datasets. DCI (Urbanek et al., 2024): 1024 summarized captions of densely captioned images. COCO (Lin et al., 2014): 1024 captions of common object images. Clotho v2 (Drossos et al., 2020): 975 audio–caption pairs from the evaluation split, one caption per clip. In total, we compare the models across WiT (image–text), DCI (image–text), COCO (image-text), AudioCaps (audio–text), and Clotho (audio–text) datasets.

Encoders supervised on language such as CLIP (vision–language) and CLAP (audio–language) show the strongest alignment with LLM representations across all prompt types, reflecting their direct training on paired data. Self-supervised models such as DINOv2, ViT-MAE, ViT-MSN, BEATs, Audio-MAE, and EAT exhibit weaker but still consistent alignment trends. BEATs+ shows higher alignment over its self-supervised counterpart, which highlights the role of additional semantic supervision in shaping compatibility with text-trained LLMs.

We also report token–alignment trends for additional language models (Figures 20, 25–27). We include Alibaba Cloud's Qwen2.5-Instruct family (Qwen et al., 2025), Meta's Llama 3.1/3.2 family (Grattafiori et al., 2024; Meta AI, 2024) and Microsoft's Phi-4 (Abdin et al., 2024), alongside our Qwen3 baselines.

These results demonstrate that our sensory-prompting findings are not tied to any specific encoder, dataset, or language model: alignment effects generalize across self-supervised and multimodally supervised encoders and across multiple LLM families (Llama 3/3.1, Phi-4, Qwen3). Moreover, the strongest alignment consistently arises from encoders with explicit multimodal supervision.

## D  VISUAL BIAS IN AUDITORY SETTING

To further understand how sensory prompts shape internal representations, we compute pairwise similarity between no cue, SEE, and HEAR generations using both centered kernel alignment (CKA) and mutual-$k$NN alignment ($k$=10) (Figure 28).

Across most scales, no cue embeddings are consistently closer to SEE than HEAR, even on the auditory AudioCaps dataset—demonstrating a persistent visual inductive bias under default prompting. To address the possibility that the instruction "imagine" may implicitly induce visual imagery, we confirm that the same trend holds under "describe" instructions (Appendix Figure 30), suggesting that auditory structure is weakly activated by default and that explicit HEAR cues greatly improve alignment with audio encoders. At 32B scale, no cue diverges from both SEE and HEAR on AudioCaps. While this may reflect a shift toward a more modality-agnostic default representations in larger models, we view this as a preliminary observation. Interestingly, this difference is less stark using "describe" instructed prompting.

We extend Figure 28 to "describe" instructed prompting in Figure 30 and to the DCI dataset in Figure 29.

## E  EXTENDED ANALYSIS OF SENSORY AXIS PROJECTIONS

To complement the qualitative distribution plots in Figure 8 (a), we quantify SEE–HEAR separation along the learned projection axis in Table 2. We report three metrics: $\Delta\mu$, the raw difference between the mean SEE and HEAR projections (larger values indicate greater directional shift); Cohen's $d$, the standardized effect size that rescales $\Delta\mu$ by within-class variance ; and AUROC, a rank-based discriminability score reflecting how well a single threshold separates SEE from HEAR (0.5 = chance, 1.0 = perfect).

Figure 31 extends our projection analysis (Figure 8 (a)) to the DCI dataset. Compared to WiT, DCI shows stronger disentanglement between SEE and HEAR cues—reflected in wider separation of

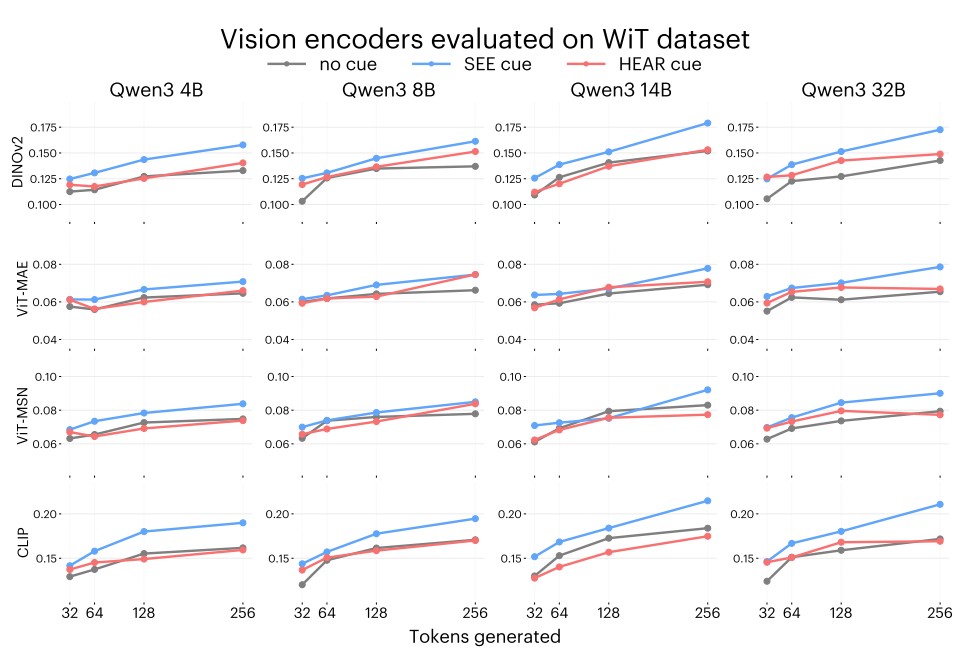

Figure 19: Extension of Figure 6 to additional sensory encoders on WIT.

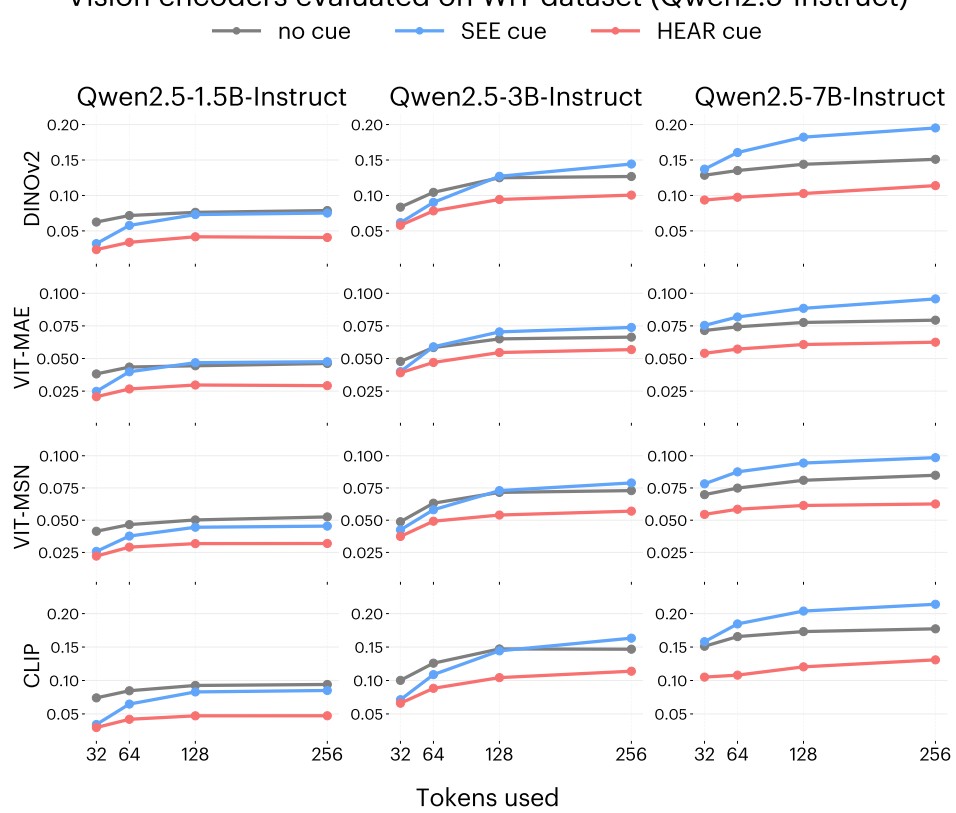

Figure 20: Extension of Figure 6 to Qwen2.5-Instruct family and additional sensory encoders on WIT .

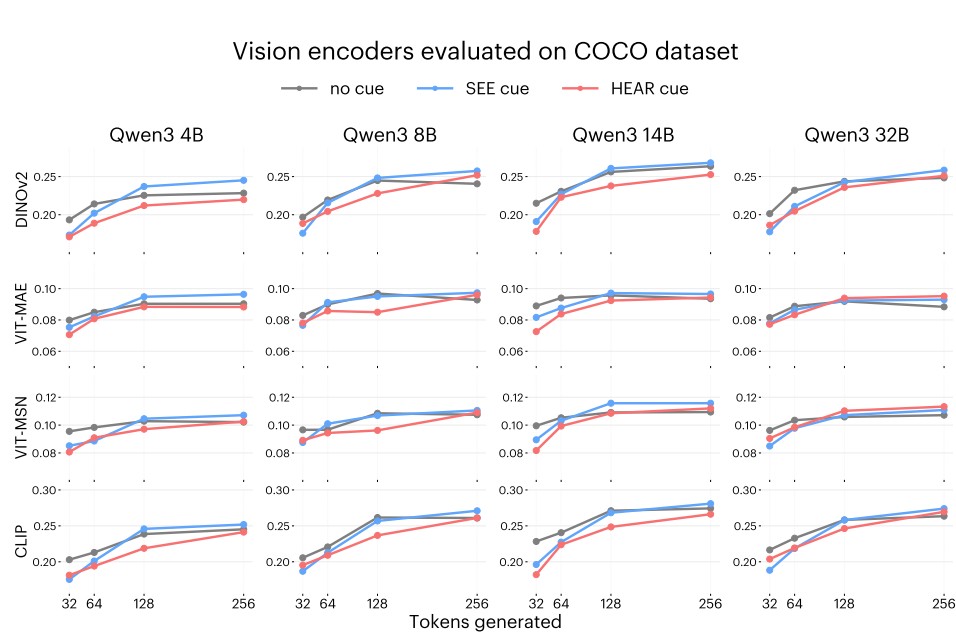

Figure 21: Extension of Figure 6 to additional sensory encoders on COCO.

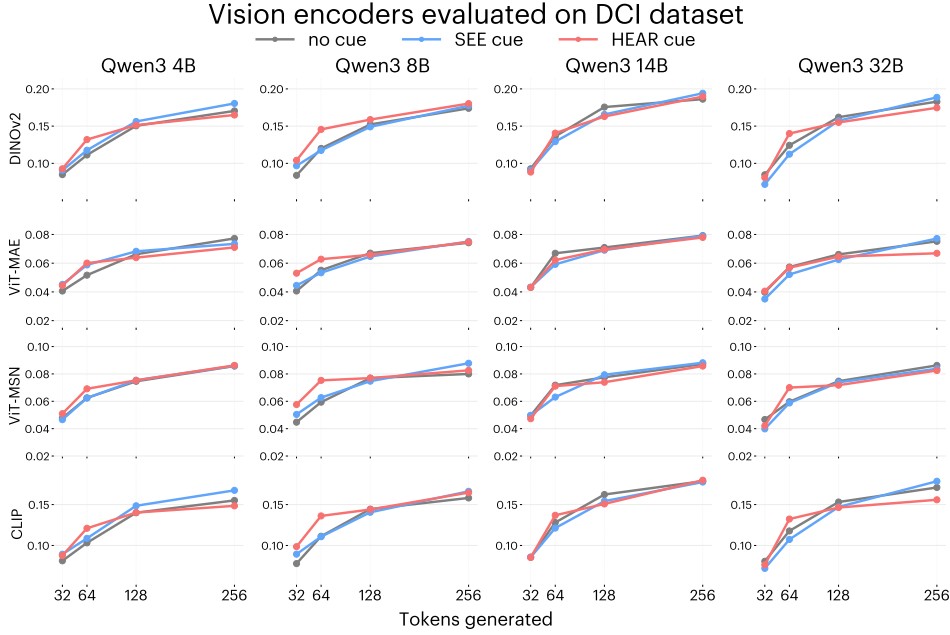

Figure 22: Extension of Figure 6 to additional sensory encoders on DCI.

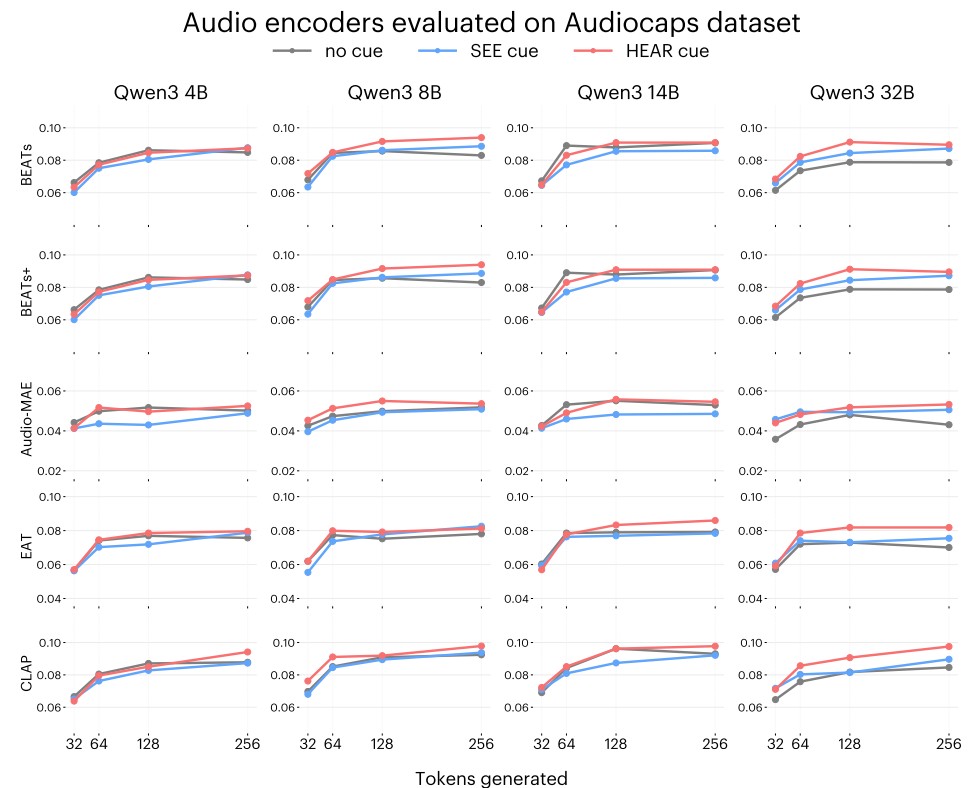

Figure 23: Extension of Figure 6 to additional sensory encoders on AudioCaps.

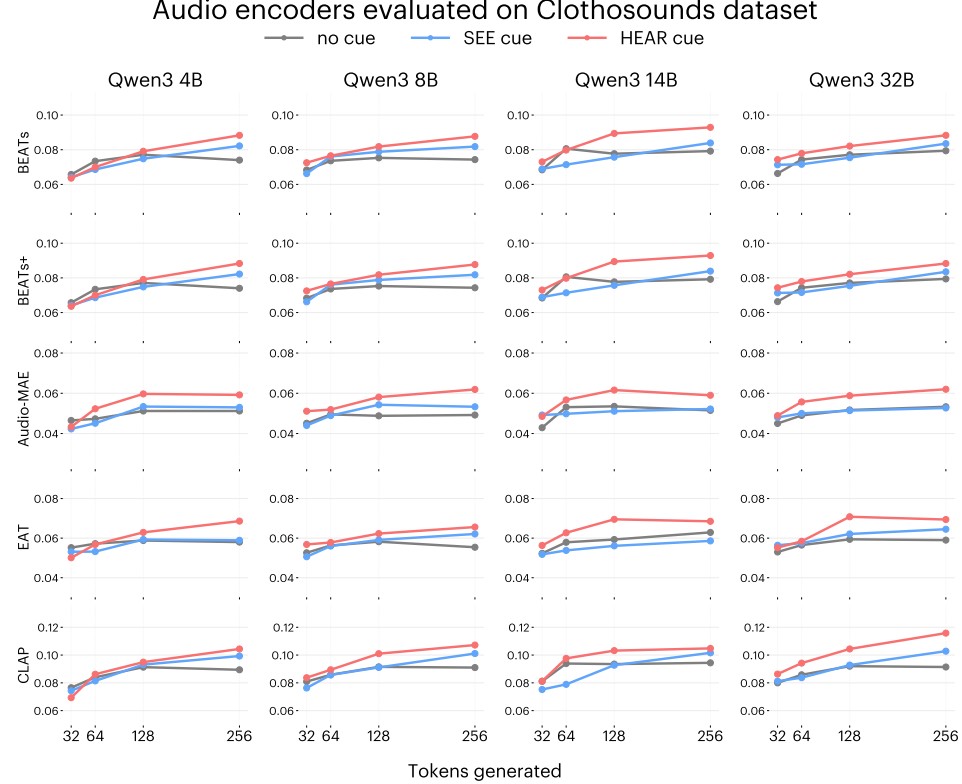

Figure 24: Extension of Figure 6 to additional sensory encoders on Clothosounds.

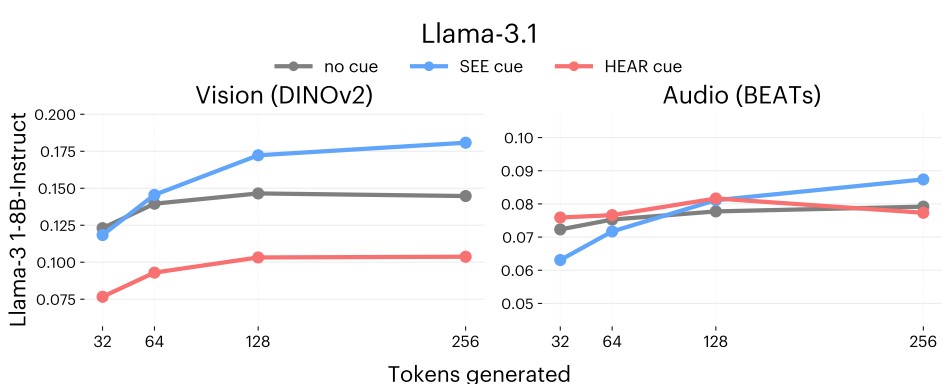

Figure 25: Extension of Figure 6 to additional language models: Llama 3.1.

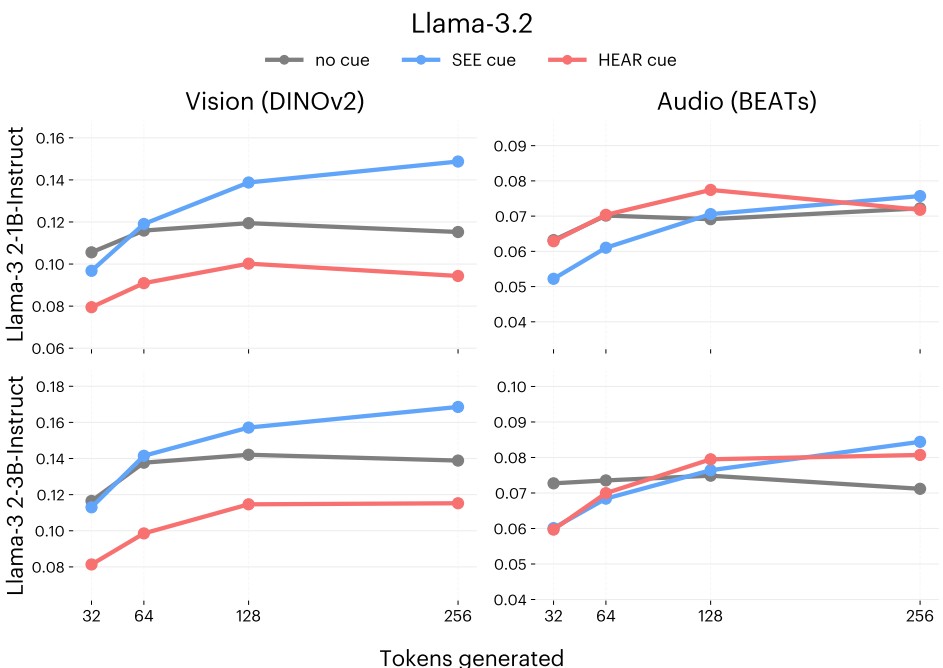

Figure 26: Extension of Figure 6 to additional language models: Llama 3.2.

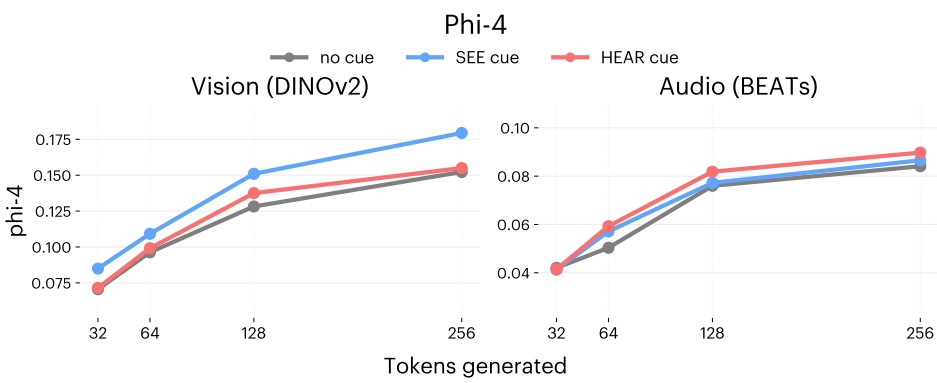

Figure 27: Extension of Figure 6 to additional language models: Phi-4.

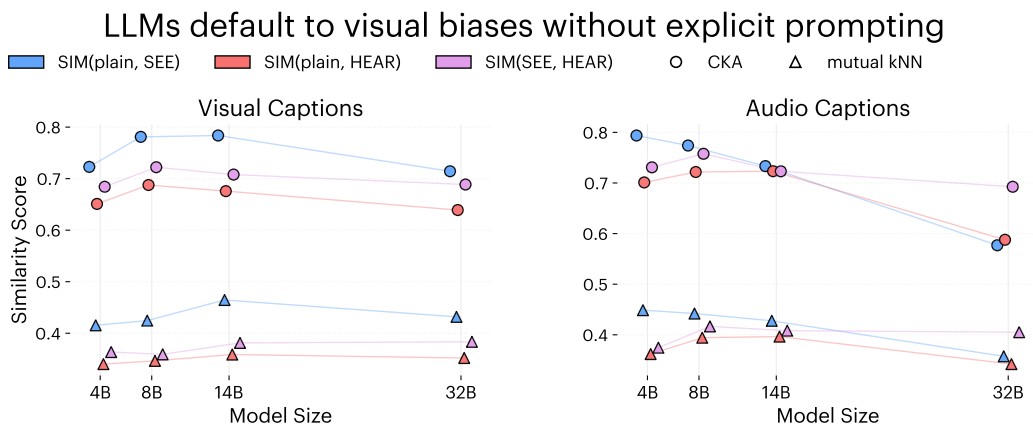

Figure 28: Similarity metrics (CKA and mutual-$k$NN) show that no cue prompts are consistently closer to SEE than HEAR—especially for audio captions—highlighting a default visual bias that diminishes with model scale.

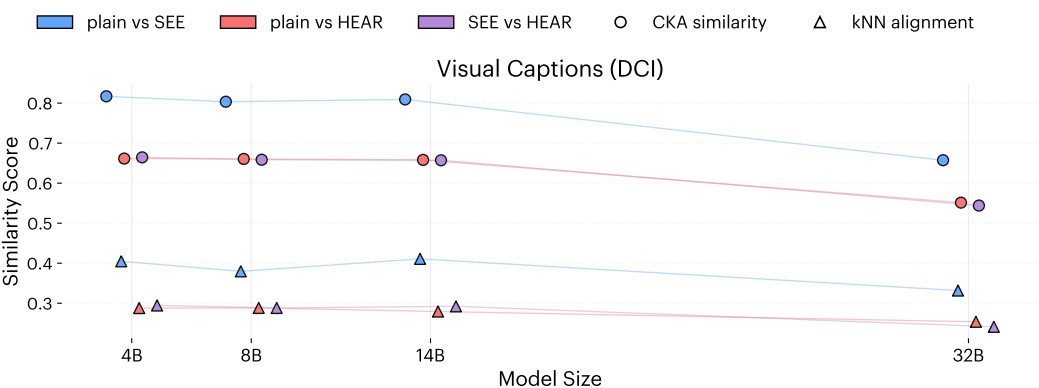

Figure 29: Extension of Figure 28 to "describe" instructed prompting.

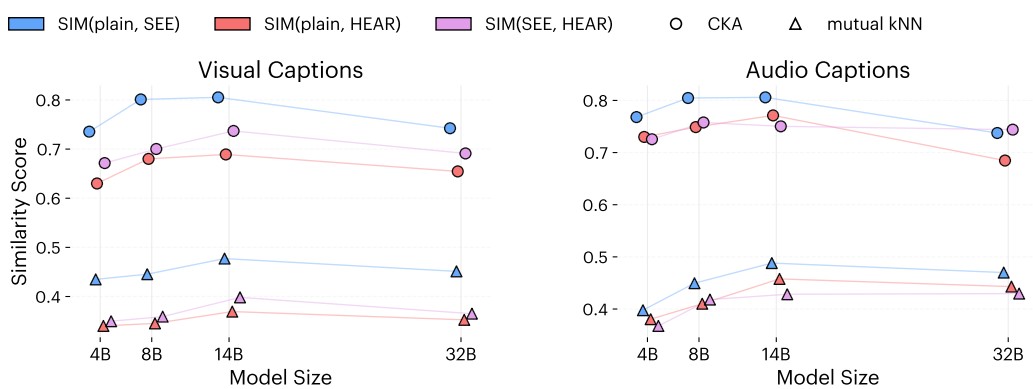

Figure 30: Extension of Figure 28 to the DCI dataset.

the projection distributions. We hypothesize this is because DCI captions contain richer inherently visual detail, such as textures, layouts, and scene composition, whereas WiT captions often reference proper nouns, locations, or events like "Finster/Nagy at the 2019 World Junior Championships" or "Unnamed hurricane of 1975 near the Pacific Northwest", which offer less explicit sensory content and may reduce the contrast in projection space.

Figure 32 shows projections under "describe" prompting, which we compare to "imagine" prompting from Figure 8 (Top). We observe that plain prompts are more evenly distributed between SEE and HEAR under the "describe" framing, suggesting that "describe" is more modality-neutral than "imagine," which may bias the model toward visual generation by default. This supports the idea that the instruction itself can influence how the model commits to a latent sensory framing, even in the absence of an explicit SEE or HEAR cue.

Table 2: Quantification of the visual–auditory disentanglement in Figure 8b.

| Dataset | Model | $\Delta\mu$ | Cohen's $d$ | AUROC |
|---|---|---|---|---|
| WiT | Qwen3 4B | 6.6 | 1.95 | 0.92 |
| WiT | Qwen3 8B | 13.3 | 2.13 | 0.94 |
| WiT | Qwen3 14B | 19.0 | 2.34 | 0.96 |
| WiT | Qwen3 32B | 21.0 | 2.65 | 0.97 |
| AudioCaps | Qwen3 4B | 10.6 | 3.14 | 0.98 |
| AudioCaps | Qwen3 8B | 19.4 | 3.10 | 0.98 |
| AudioCaps | Qwen3 14B | 28.8 | 4.11 | 0.99 |
| AudioCaps | Qwen3 32B | 32.4 | 4.52 | 1.00 |

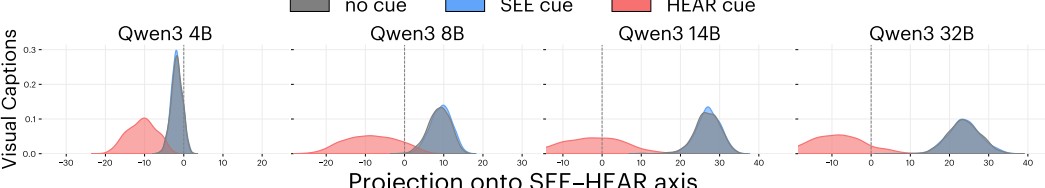

Figure 31: Extension of Figure 8 (Top) to DCI dataset.

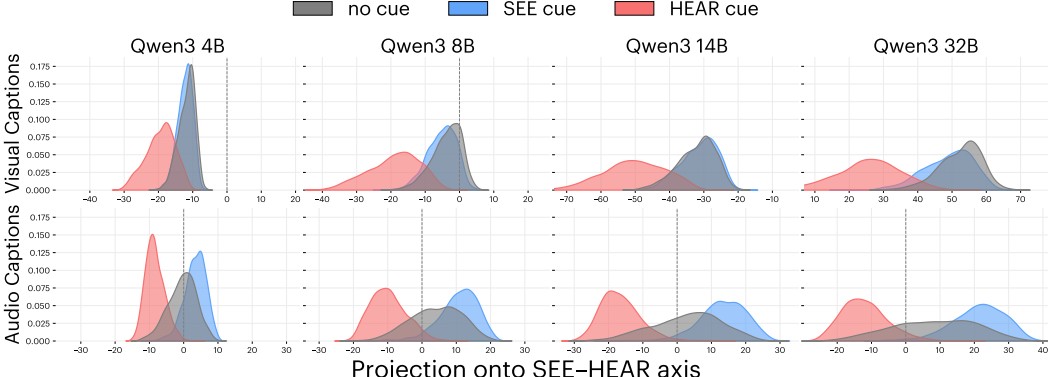

Figure 32: Extension of Figure 8 (Top) to "describe" instructed prompting.

## F EXTENDED ANALYSIS OF VQA IN TEXT SPACE

We evaluate caption-based visual question answering on the MME benchmark (Fu et al., 2023a), following the "VQA without V" setup of Chan et al. (2025). Each image in MME is first converted into a natural language caption using Qwen2.5-VL-3B-Instruct. To ensure captions remain faithful to the source modality, we use a category-specific prompting strategy during captioning: for categories involving code or mathematics, the model is instructed to *transcribe line-by-line without interpretation*; for all other categories, the model is instructed to *describe only visible characteristics without interpretation or commentary*. This prevents the captioner from injecting additional semantic reasoning that would not be directly available from the image itself.

Given these captions, we then construct (caption, Q) pairs and evaluate Qwen3-14B as the answering model under two different prompting conditions:

- No cue: `You will be given a CAPTION and a question.  Your role is to answer the question only with 'yes' or 'no' by using the CAPTION. CAPTION: {caption}`

- SEE cue: `You will be given a CAPTION and a question.  Your role is to answer the question only with 'yes' or 'no' by imagining what it looks like to see the CAPTION. CAPTION: {caption}`

The key manipulation is the addition of the *visual framing cue* ("imagine what it would look like to see..."), which biases the model toward perceptual simulation when interpreting the caption. Both conditions are evaluated across all categories of the MME benchmark except OCR, which inherently requires direct text recognition from images.

In the following, we present an example drawn from the MME dataset. The example shows the original image, the generated caption, the associated yes/no question, and the generations from Qwen3-14B under both cue conditions. These examples illustrate the mechanism by which sensory prompting allows the model to answer correctly by invoking a visual imagination of the scene's text caption.

### F.1 EXAMPLE: ARTWORK/10256.JPG

Q: [Y] Does this artwork exist in the form of painting?

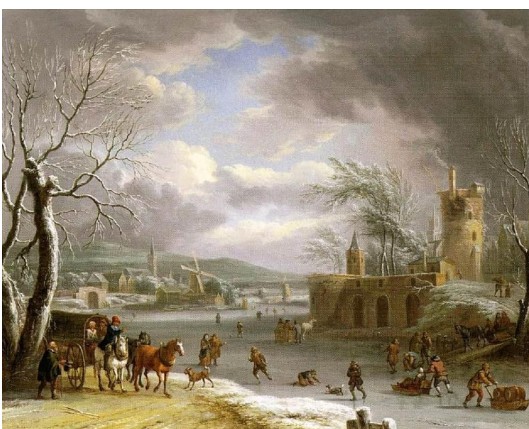

Figure 33: Caption: A winter scene with people skating on a frozen river, horses pulling carriages, and buildings in the background under a cloudy sky.

---

**No cue output (incorrectly answers [N])**

Okay, let's see. The user provided a caption describing a winter scene with people skating, horses pulling carriages, buildings under a cloudy sky. The question is whether this artwork exists as a painting. The answer needs to be yes or no based on the caption.

First, the caption itself is a description of a scene. The question is about whether there's a painting that depicts this. But the caption doesn't mention anything about a painting. It just describes the scene. So, the caption is likely a description of a painting, but the existence of the artwork as a painting isn't confirmed by the caption. The caption could be from a photograph, a drawing, or even a written description. Without explicit mention of it being a painting, I can't assume it's a painting. Therefore, the answer would be no because the caption doesn't state that it's a painting. The user might be testing if I can infer the medium from the description, but since it's not specified, the answer is no.

---

**SEE output (correctly answers [Y])**

Okay, let's see. The user provided a caption describing a winter scene with people skating, horses pulling carriages, buildings under a cloudy sky. The question is whether this artwork exists as a painting. They want a yes or no answer.

First, I need to figure out if the caption is referring to a real painting. The caption sounds like a description that could be from a well-known painting. The elements mentioned—frozen river, skaters, horse carriages, buildings, cloudy sky—are classic elements found in 19th-century landscape paintings.

Wait, the most famous painting that comes to mind is "Winter Landscape with Skaters" by Hendrick Avercamp. He was a Dutch painter known for his winter scenes. That painting does have people skating on a frozen river, horse-drawn carriages, and buildings in the background with a cloudy sky. So if the caption is describing that painting, then the answer would be yes.But how do I know if the caption is referring to that specific painting? The user didn't mention the title or artist. However, the elements are very specific and match Avercamp's work. Since the question is whether the artwork exists as a painting, and given that such a painting does exist, the answer would be yes.

Alternatively, maybe the user is asking if the described scene is a real painting, not just a hypothetical one. Since the description matches a known painting, the answer is yes. If there's no such painting, the answer would be no, but in this case, there is. So the answer should be yes.

# G  FULL PROMPTED GENERATION EXAMPLES

All generations in this section originate from the following caption (WiT):

```
Awaiting a pitch -- batter, catcher, and umpire in
baseball
```

## G.1  128-TOKEN TEXT GENERATIONS

Below are full 128-token text generations from Qwen3-32B when prompted with the template outlined in Figure 3.

---

**SEE output (128 token generation)**

Okay, the user wants me to imagine what it looks like to see a baseball scene where the batter, catcher, and umpire are waiting for a pitch. Let me break this down. First, I need to visualize a baseball field, specifically the area around home plate. The batter is up to bat, so they're standing in the batter's box, maybe adjusting their grip on the bat or their stance. The catcher is behind the plate, wearing all that gear—helmet, chest protector, knee pads. They're probably crouched low, watching the pitcher closely. The umpire is standing near home plate,

---

**HEAR output (128 token generation)**

Okay, the user wants me to imagine what it would sound like to hear the moment before a pitch in baseball, involving the batter, catcher, and umpire. Let me break this down. First, I need to think about the typical sounds in a baseball game during that specific moment. The atmosphere in the stadium, the crowd's reactions, the communication between players, and any equipment sounds.

Starting with the batter: they might be adjusting their batting gloves, maybe tapping the plate with their foot or the bat. There's the sound of the bat being raised, maybe some deep breathing. The batter might also be listening to the

---

**No cue output (128 token generation)**

Okay, so the user wants me to imagine a scenario in baseball where a batter, catcher, and umpire are involved, specifically awaiting a pitch. Let me break this down. First, I need to visualize the setup. The batter is at the plate, ready to hit. The catcher is behind home plate, crouched down, signaling the pitcher. The umpire is positioned behind the batter, right? Or maybe there are multiple umpires, but the main one here is the home plate umpire.

I should think about the sequence of events. The pitcher is about to throw the ball. The batter is focused,

---

## G.2  SENSORY REDIRECTION GENERATIONS

Below are full 128-token text generations from Qwen3-32B when prompted with the sensory redirecting template outlined in Figure 7, using outputs from Appendix G.1.

---

**SEE-to-HEAR output (128 token generation)**

The scene hums with anticipation, the only sounds the soft rustle of grass underfoot and the occasional murmur of the crowd in the distance. A tense silence hangs in the air as the batter shifts their weight, the creak of leather as they adjust their grip on the bat. The catcher's gear creaks and shifts as they crouch low behind the plate, their breath steady but audible, the faint clink of their mask as they glance toward the pitcher. The umpire stands rigid, his voice low and firm as he calls the count, each word crisp and clear, cutting through the quiet. Then, a sudden

---

> **HEAR-to-SEE output (128 token generation)**
>
> The scene is tense and electric, frozen in the instant before the pitch. The batter stands at the plate, gripping the bat tightly, his muscles coiled with anticipation. His eyes are locked onto the pitcher, scanning for the slightest movement. The catcher crouches behind the plate, gloved hands poised, his body ready to react. His eyes are on the umpire, waiting for the signal that will set the play in motion. The umpire stands at the center of the action, one hand raised, his posture rigid with authority. Around them, the stadium hums with quiet intensity—fans leaning forward in their seats, the

### G.3 SENSORY ABLATION GENERATIONS

Below are full 128-token text generations from Qwen3-32B when prompted with the sensory ablation prompt in Figure 7, using outputs from Appendix G.1.

> **Sensory ablation of SEE output (128 token generation)**
>
> Okay, the user wants me to imagine a scene involving a baseball game where three individuals are present and waiting for an action to occur. First, I need to picture a setting related to a baseball game, specifically near the area where the batter stands. One person is positioned to take their turn, possibly preparing for an action. Another individual is located behind a designated area, wearing protective equipment, and is likely observing another person closely. A third person is positioned nearby, ready to oversee the action.

> **Sensory ablation of HEAR output (128 token generation)**
>
> Okay, so I need to imagine what it would be like to experience the moment when a batter, catcher, and umpire are waiting for a pitch in baseball. Let me break this down. First, I should recall what happens during that specific moment in a baseball game. The batter is at the plate, the catcher is behind the plate, and the umpire is standing near the plate as well. The pitcher is about to throw the ball.
>
> What is happening in that scene? Let me think. There is general activity around the area—maybe people speaking, moving, or making noise. The catcher might be communicating with the pitcher,

> **Sensory ablation of no cue output (128 token generation)**
>
> The user is referring to a situation in a game where a pitch is about to be thrown, involving three key participants: the batter, the catcher, and the umpire. Each has a specific function in the sequence of events that occur during this moment.
>
> The batter is positioned to attempt to make contact with the ball. They observe the pitcher's actions, interpret signals, and prepare to react based on their assessment of the incoming pitch.
>
> The catcher is located behind the base where the batter is facing. They coordinate with the pitcher, prepare to receive the ball, and may take actions to influence the outcome of the pitch. They also have a

## G.4  512-TOKEN TEXT GENERATIONS

Below are full 512-token text generations from Qwen3-32B when prompted with the template outlined in Figure 3. This particular example extends the first sample in Figure 4.

---

**SEE output (512 token generation)**

Okay, the user wants me to imagine what it would look like to see the scene where a batter, catcher, and umpire are waiting for a pitch in baseball. Let me break this down. First, I need to visualize a baseball field. The batter stands at the home plate, right? They're holding the bat, probably in a ready position, feet shoulder-width apart. Their body language should show they're focused, maybe squinting at the catcher or the umpire.

The catcher is behind the plate, wearing all that gear—helmet, chest protector, knee pads. They're crouched down, signaling to the pitcher. Their gloves are up, ready to catch the pitch. Maybe they're communicating with the pitcher through hand signals or verbal cues. The mitt is essential here, big and visible.

Then there's the umpire. They're standing near the plate, maybe a few steps back. They have the plate behind them, and they're wearing the blue uniform with the mask and chest guard. Their hands are probably up, ready to call balls and strikes. Their expression is intense, focused on the pitcher's release point to make accurate calls.

I should also think about the setting. The sun might be setting, casting long shadows, adding a dramatic effect. The field is green, the infield dirt, the outfield grass. Maybe some spectators in the stands to give context. The atmosphere is tense, the quiet before the action. The pitcher is off to the side, maybe adjusting the ball or getting ready to throw.

I need to make sure I capture the details of each role. The batter's stance, the catcher's signals, the umpire's positioning. Also, the equipment: bats, gloves, uniforms. Maybe the angle of view. Should I describe from a spectator's perspective? Maybe a wide shot to include all three, then close-ups on each. Highlight the anticipation in their expressions and body language.

---

**HEAR output (512 token generation)**

Okay, the user wants me to imagine what it would sound like to hear the moment before a pitch in baseball, involving the batter, catcher, and umpire. Let me break this down.

First, I need to recall the typical sounds and atmosphere in a baseball game during that specific moment. The scene is the pitcher's mound, with the batter waiting, the catcher behind the plate, and the umpire nearby. What sounds are present?

There's the general ambient noise of the stadium: crowd murmurs, maybe some distant cheers or chatter. The crack of a bat from another field if it's a doubleheader or a minor league game nearby. The sound of players moving, like cleats scraping on the dirt, gloves being adjusted. The umpire might be talking to himself or the catcher, maybe adjusting his mask or gear. The catcher could be signaling to the pitcher, so there might be hand signals and maybe some quiet communication, though not loud. The batter might be tapping the plate, adjusting his gloves, maybe muttering to himself or listening to the umpire.

Wait, the user specified "awaiting a pitch," so the moment just before the pitcher delivers. The pitcher is on the mound, wind-up or stretch position. The tension is high. What sounds are there in that split second? The umpire's focus, maybe breathing heavily if it's a tense game. The catcher's gear might make some noise as he positions himself. The batter's focus, maybe some last-second adjustments. The crowd might be quiet, holding their breath, or murmuring excitedly.

Also, specific sounds like the umpire's "Play!" or "Batter up!" if that's part of the sequence. But during the actual awaiting moment, it's more about the silence or the taut atmosphere. Maybe the creak of equipment, the sound of the pitcher's uniform moving, the tension in the muscles as they prepare.

I should also consider the acoustics of a baseball stadium. Open-air, so sounds carry but also get absorbed. The distance between the catcher, batter, and umpire affects how their sounds are perceived. The umpire might be near the plate, so his movements are close to the batter and catcher.

Possible sounds: the umpire's voice giving a subtle cue, the catcher's mitt moving, the batter's bat twirling or adjusting. Maybe the sound of the pitcher's foot hitting the rubber. The tension in the air, almost a silence before the action. The crowd's noise might swell in the background just before the pitch.

---

**No cue output (512 token generation)**

Okay, the user wants me to imagine a scenario in baseball where a batter, catcher, and umpire are waiting for a pitch. Let me break this down. First, I need to visualize the setup. The batter is at the plate, ready to hit. The catcher is behind the plate, signaling the pitcher. The umpire is positioned behind the plate as well, right? Wait, no, the home plate umpire is there, but there are other umpires too. Maybe the user is focusing on the main three: batter, catcher, and home plate umpire.

So, what's happening here? They're all waiting for the pitch. The pitcher is about to throw. Let me think about each person's role. The batter is trying to anticipate the pitch type, location, and speed. The catcher is setting up, maybe ready to block the plate if needed, and also communicating with the pitcher. The umpire is focused on calling balls and strikes, but also watching for other things like if the pitch is a strikeout or if there's a hit.

Wait, but the user mentioned "awaiting a pitch," so maybe it's before the pitch is thrown. The batter is in the box, the catcher is in position, and the umpire is in place. The pitcher is on the mound, getting ready. The tension is high. The batter is crouched, hands in position, eyes on the pitcher. The catcher is signaling, maybe with a hand signal or a sign from the dugout. The umpire is standing behind the plate, hands at the sides, ready to call the game.

I should describe the atmosphere. The crowd might be silent, waiting for the pitch. The batter is focused, maybe taking a deep breath. The catcher is focused on the pitcher's windup, ready to react. The umpire is also watching the pitcher's motion for any sign of a balk or illegal motion. The count could be anything—maybe a close count like 3-2, which adds tension. The outcome of this pitch could determine the at-bat, maybe even the game.

I need to make sure I cover each character's perspective. The batter's anticipation, the catcher's communication and readiness, the umpire's role in calling the strike zone. Also, the physical positions: batter in the box, catcher behind the plate, umpire behind the plate but maybe a step back? Or is the umpire right behind? I think the home plate umpire stands right behind the

# H VISUAL PROMPTING IMPROVES PERCEPTUAL GROUNDING IN IMAGE GENERATION

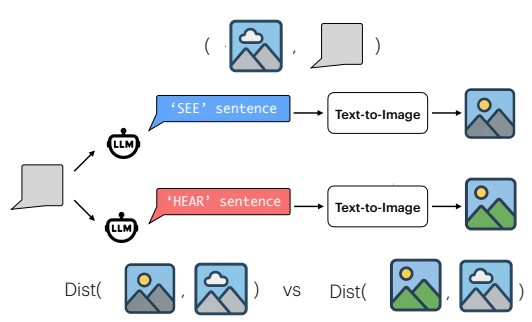 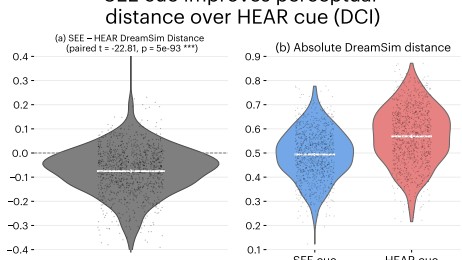

(a) Experimental setup: LLM generates a cue-conditioned sentence, which is passed to a diffusion model.

(b) SEE-prompted captions yield lower Dream-Sim distance to reference images (better visual grounding).

| SEE prompt | Write a sentence in the format \<sentence\>\</sentence\> to describe the visual scene of: {caption} |
| HEAR prompt | Write a sentence in the format \<sentence\>\</sentence\> to describe the sound of: {caption} |

(c) Prompt templates used to elicit a single declarative sentence for the diffusion model.

Figure 34: Effect of sensory prompting on visual fidelity in diffusion-based image generation. (a) Experimental setup. (b) SEE cues improve visual grounding compared to HEAR cues, as measured by DreamSim (lower is better). (c) Prompt templates for cue-conditioned sentence generation.

To test whether sensory cues influence downstream behavior, we generate sensory-cued captions of images and pass them to Stable Diffusion XL (Podell et al., 2023). Resulting generated images are compared to originals using DreamSim (Fu et al., 2023b), a perceptual similarity metric. Lower DreamSim scores indicate more faithful reconstructions.

We prompt Qwen3-32B with SEE or HEAR using the templates in Figure 34. The model generated a single sentence describing a scene, which was then passed to Stable Diffusion XL to produce an image.

We compared the generated images to ground-truth images using DreamSim, a learned perceptual similarity metric (see Figure 35 and Figure 37 for qualitative examples). Captions produced under the SEE cue consistently yielded more visually faithful generations. Outputs prompted with SEE often closely resembled the original captions, suggesting that the model recognizes their inherently visual nature and preserves that framing when instructed.

These findings demonstrate that even a minimal cue can shift both the model's internal representations and its assumptions about the sensory context underlying the text—ultimately influencing what information is emphasized in downstream outputs.

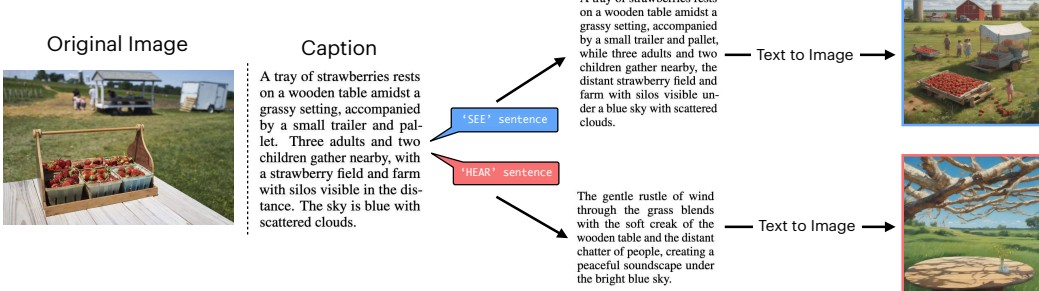

Figure 35: Illustrative example of the text-to-image generation pipeline from Figure 34 (Left). Starting from the same input caption and reference image, the LLM generates a single sentence under either a SEE or HEAR cue. The SEE-conditioned sentence closely mirrors the original caption and emphasizes visual layout and concrete scene elements, while the HEAR version shifts toward ambient auditory details. Each description is then passed to Stable Diffusion XL to generate an image. The resulting images reflect the modality-specific focus induced by the prompt cue.

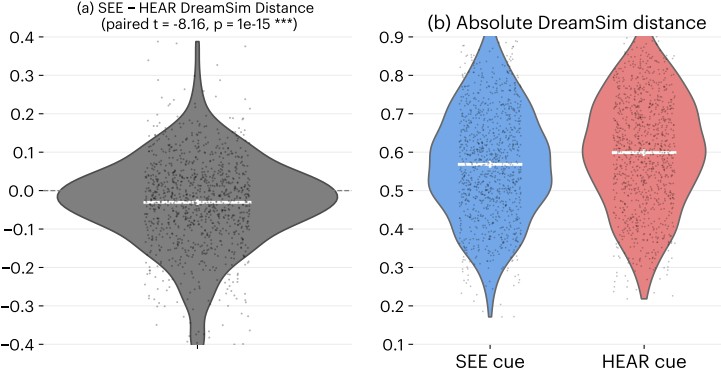

Figure 36: Extension of Figure 34b to WiT dataset.

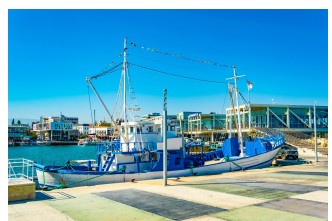 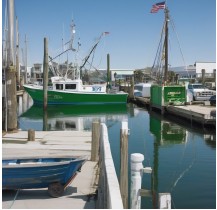 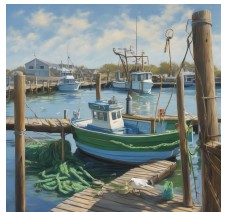

(a) A blue and white fishing boat is docked in a small harbor, surrounded by commercial buildings. The boat has two white masts with fluttering flags and is secured with green ropes. A green pickup truck sits nearby. The dock is made of alternating white, green, and blue concrete blocks, creating a unique pattern. The sky is clear and blue.

(b) A blue and white fishing boat is docked in a small harbor, surrounded by commercial buildings, its two white masts adorned with fluttering flags and secured with green ropes, while a green pickup truck sits nearby on a dock made of alternating white, green, and blue concrete blocks beneath a clear and blue sky.

(c) The creaking of the green ropes, the soft lapping of waves against the wooden dock, and the distant chatter of seagulls fill the quiet harbor as the blue and white fishing boat sways gently in the clear blue sky.

Figure 37: (a) Ground-truth image. (a) Generation from a SEE-cued prompt. (c) Generation from a HEAR-cued prompt.

