# OpenReview forum: "Words That Make Language Models Perceive"
_ICLR.cc/2026/Conference — ICLR 2026 Conference Desk Rejected Submission_

### Official Review · Reviewer_t8zj · 2025-10-23

**Soundness:** 2
**Presentation:** 4
**Contribution:** 3
**Rating:** 6
**Confidence:** 4

**Summary:**

Building on the platonic representation hypothesis, this article explores how the cosine similarity between a text model's representation on one side and either a vision or an audio model's representation on the other side can be influenced by prompting. In particular, the authors discover that when the prompt encourages the model to describe visual aspects, then similarity to the vision model increases, while if it's asked to describe auditory aspects, then similarity to the auditory model increases.

**Strengths:**

1. The research question is well motivated an interesting (to me, but also likely to others that are interested in the platonic representation hypothesis and related work)
2. Great figures, very helpful and also beautifully designed
3. The view of alignment not as a fixed property (as in e.g. the platonic representation hypothesis paper and most of the literature), but as something that can be influenced through prompting, is an important and valuable perspective (and has parallels to e.g. prompt steerability [Miehling et al. 2025, "Evaluating the Prompt Steerability of Large Language Models"], biases in VLMs [Gavrikov et al 2025, "Can we talk models into seeing the world differently"]).
4. The analysis provides a framework that could be expanded to SSL vs. supervised encoders and VLMs as well.

**Weaknesses:**

Major:

1. The magnitude of the observed effects is small. While the results are systematic and I don't doubt their validity (the core hypothesis is supported by appropriate evidence), a cosine similarity increase from e.g. 0.14 to 0.16 is just a bit small to make big claims. In order to put the results into perspective, it would help to put the alignment scores into context - e.g., could a lower and upper bound be computed? Akin to "explained variance", could a percentage be calculated w.r.t. these lower and upper bounds in terms of how much variance in e.g. a visual representation is captured by the text representation?

2. Relatedly, and as a consequence: The title - "words that make language models perceive" - is catchy but also feels like a bit of a stretch. Yes there are some words that increase an LLM's cosine similarity with a perception model from e.g. 0.14 to 0.16, but a more accurate description would be "prompts that make language model representations slightly more aligned with perception model representations". In light of the small magnitude of the results, I strongly recommend that the authors make a pass throughout the paper to contextualize claims accordingly and tone some of them down a bit - it's a solid investigation and the paper doesn't need to make big claims based on small effect sizes.

3. The scientific contribution (as in providing knowledge / discovery) is there and that's great; however, the paper's impact could be strengthened by showing that those insights can be turned into downstream benefits of some sort. While a "VQA without V" tasks is explored in Section 3.6, this task is highly artificial: it's appropriate for the experiment purpose, but doesn't demonstrate credible downstream benefit. One possible way to show the effect on downstream evals could be to add an "alignment penalty" or "alignment encouragement" term to training a language model, and testing how this influences downstream task performance. Just one thought, other options are possible as well, but it's always nice to see a paper ending without a reader thinking "oh nice, but also, so what - how does this knowledge now help me?". Please feel free to ignore this point if it's not something you're aiming to achieve with this paper though, a pure scientific contribution is of course valid and valuable in itself - just thinking of ways to broaden the paper's possible impact.

Minor:
- Code not provided (the authors promise to provide it later)

I'm very motivated to increase my score if my concerns can be addressed.

**Questions:**

- for mutual kNN, what's chance alignment?
- line 51: "more similar" - compared to what? Suggesting to add context
- line 76: can "perceptually grounded" be defined?
- line 89: "VQA in text modality" doesn't make much sense at first glance (unless one reads the rest of the paper), could be paraphrased/explained in the results summary to make it more self-contained
- Figure 2: suggesting to add explanation for alignment to what/which system in the figure caption.
- is Figure 5 a randomly selected or cherry-picked example? Either is fine, suggesting to state the selection method in the caption.
- line 415: can this be confirmed statistically?
- line 463: why "interpretable"?
- line 467 "an LLM can act like an image or audio encoder" - I disagree; people expect something very different when reading this (a model that actually takes in image or audio as input). I get what you're trying to say but suggesting to rephrase to make this distinction clear.

---

> ### Author Response · Authors · 2025-11-18
>
> Thank you for taking the time to write a thoughtful review! We are encouraged that you found our paper well motivated, interesting, and an important perspective in the context of representation alignment literature. We also appreciate that you found our figures helpful and our analysis applicable to other topics of interest. We kindly respond to your concerns and number them for convenience:
>
> # 1.
> > it would help to put the alignment scores into context - e.g., could a lower and upper bound be computed?
>
> > for mutual kNN, what's chance alignment?
>
> The expected chance alignment is $\approx k/n$. In our setup with $k=10$ and $n=1024$, this predicts chance alignment $\approx 10/1024 = 0.0098$. We estimate this empirically by permuting the pairing between Qwen3-14B (128 tokens, SEE cue) and DINOv2 embeddings and recomputing mutual-kNN alignment over 20 permutations, which yields a chance alignment of  $0.0097 \pm 0.0001$ (mean $\pm$ SE).
>
> As a loose upper bound, we also report m-kNN between different vision encoders and audio encoders.
>
> Vision models:
> |   | clip-vit-base-patch32 | dinov2-base | vit-mae-base | vit-msn-base |
> | --------------------- | --------------------- | ----------- | ------------ | ------------ |
> | clip-vit-base-patch32 | 1.0   | 0.476  | 0.273   | 0.259|
> | dinov2-base   | 0.476| 1.0 | 0.336   | 0.3433  |
> | vit-mae-base  | 0.273| 0.336  | 1.0  | 0.332   |
> | vit-msn-base  | 0.259 | 0.343  | 0.332   | 1.0  |
>
>
> Audio models:
> |   | BEATs_iter3 | BEATs_iter3_plus_AS2M | EAT-base_epoch30_pretrain | audiomae   | clap-htsat-fused |
> | ------------------------- | ----------- | --------------------- | ------------------------- | ---------- | ---------------- |
> | BEATs_iter3   | 1.0 | 1.0   | 0.381| 0.300 | 0.250   |
> | BEATs_iter3_plus_AS2M | 1.0 | 1.0   | 0.381 | 0.300 | 0.250   |
> | EAT-base_epoch30_pretrain | 0.381  | 0.381| 1.0   | 0.279 | 0.289   |
> | audiomae  | 0.300  | 0.300| 0.279 | 1.0| 0.176   |
> | clap-htsat-fused  | 0.250 | 0.250| 0.289 |  0.176 | 1.0  |
>
> # 2.
> > In light of the small magnitude of the results, I strongly recommend that the authors make a pass throughout the paper to contextualize claims accordingly
>
> Thank you for the suggestion! We have revised the paper accordingly.
>
> We also want to clarify these numbers in the context of prior work. In our paper, generative representations increase LLM–vision alignment relative to single-pass embeddings (e.g., 0.12 -> 0.14), and adding sensory prompting yields an additional shift (e.g., 0.14 -> 0.16).  In the original PRH paper [1], Figure 3 shows mutual-kNN alignment between DINOv2 and language models ranging from 0.56B to 65B parameters; across this entire scale, alignment increases from roughly 0.10 to 0.16. In follow-up work [2], Figure 10 reports changes on the order of 1e-3 when varying the amount of structured reasoning data in pretraining. In this light, the inference-time shifts we observe for a single, fixed model across prompt conditions are even comparable to training-time interventions that have been studied.
>
> We acknowledge that these are still small absolute effect sizes, and have clarified our limitations.
>
> # 3.
> > the paper's impact could be strengthened by showing that those insights can be turned into downstream benefits of some sort.
>
> Thank you for the suggestion! We agree, and we acknowledge that turning alignment signals into training objectives that improve standard benchmarks is still an active and open area of work. For example, in spirit to your suggestion of “alignment encouragement”, [3] reports modest downstream gains when aligning language model representations to a vision model during training. More broadly, we note that sensory prompting is not straightforward to map onto standard downstream tasks, because in practical settings one might simply use a multimodal model instead.
>
> # 4.
> > Code not provided (the authors promise to provide it later)
>
> We have anonymized the codebase for your convenience: https://anonymous.4open.science/r/sensory.
>
> # 5.
> > line 415: can this be confirmed statistically?
>
> In the revision, we now evaluate three Qwen-family models (Qwen3-14B, Qwen2.5-Instruct-7B, Qwen2.5-Instruct-14B) and include standard error.
>
> # 6.
> We have addressed the other phrasing and clarity concerns in the resubmitted manuscript.
>
> —
>
> Thank you again for your helpful feedback! Do these answers address your concerns with the paper? If not, what further clarification or modifications could we make to improve your score?
>
> References:
> - [1] Huh, Minyoung, et al. "The platonic representation hypothesis." _Forty-first International Conference on Machine Learning_ (2024).
> - [2] Han, Junlin, et al. "Learning to see before seeing: demystifying LLM visual priors from language pre-training." _arXiv preprint arXiv:2509.26625_ (2025).
> - [3] Gan, Yulu et al. “Cross-modal alignment regularization: enhancing language models with vision model representations.” _Second Workshop on Representational Alignment at ICLR 2025_ (2025).

---

> > ### Comment · Reviewer_t8zj · 2025-11-28
> > **Thank you - some questions about how rebuttal is reflected in article remain**
> >
> > I'd like to thank the authors for their rebuttal and for providing lower/upper bounds and the code. With respect to showing benefits on downstream tasks - acknowledged that this is not what you're aiming for.
> >
> > In terms of the rebuttal, I still have a few questions if you don't mind (based on my original review):
> >
> > 1. What's the percentage of the observed effects w.r.t. these lower and upper bounds? Are you plotting this in the revised paper (if so, where), or have you decided not to incorporate this context in the article? (Generally, my questions & review aim at improving the manuscript; thus I'm interested in hearing not just the answer but also whether the answer is now present in the manuscript itself, given that the article is what I'm asked to assess as a reviewer, and it's also what future readers will see.)
> >
> > 2. Related to weakness #2, which statements have been revised, how, and where? For instance, in your rebuttal you write "we acknowledge that these are still small absolute effect sizes"; in the limitations section I couldn't find such a statement, only "we have not fully explored the degree to which this alignment can be improved" which has a different meaning.

---

> > > ### Author Response · Authors · 2025-11-30
> > >
> > > Thank you for the questions!
> > >
> > > On the "percentage of the observed effects w.r.t. lower and upper bounds," we are not sure what specific percentage is being requested. In the manuscript, we contextualize the effect sizes by comparing them to prior work. In the limitations section we note that the alignment we observe is "at a scale comparable to Huh et al. (2024)" and that "LLM alignment to audio encoders is lower than alignment to vision encoders and less reliably steerable." We agree that developing a normalized notion of effect size relative to such bounds could be useful for future work, since it would apply to kernel alignment methodology more broadly.

---

### Official Review · Reviewer_875U · 2025-10-28

**Soundness:** 2
**Presentation:** 2
**Contribution:** 2
**Rating:** 2
**Confidence:** 2

**Summary:**

The paper investigates the hypothesis that text-only LLM can be steered towards aligning their internal representations with vision or audio encoders by *sensory prompting*. The authors define *sensory prompting* as text-prompts asking the model to *see* or *hear* in context of a next token prediction setting. The alignment between different embedding spaces of LLMs and vision or audio encoders is measured via cosine-distance kernels over auto-regressive input sequences. Using this measure, the authors claim to show the ability to align text-only LLMs with representations of other modalities in a series of experiments.

**Strengths:**

In general, the question of models trained on different (or mixed) modalities represent the information extracted from the training data and how these representations could be linked/transferred is of high theoretical interest and practical impact. Hence, the paper raises a valid and interesting question.

**Weaknesses:**

Unfortunately, the paper suffers from three main weaknesses: a) unclear technical description of the alignment measurement (this part probably can be fixed in a revision) and b) highly inconclusive experimental results which do NOT provide sufficient evidence for the proposed hypothesis. Finally, c) the paper ignores the likely high influence of the LLM training data.

a) section 2.1 and 2.2 which describe the generation of of the model representation vectors and the alignment measure are very hard to follow. While the description of the computation of $z_g^{(p)}$ lacks many details and would definitely benefit from a visualization in a figure, it remains unclear how it is actually used in 2.2 and how the intersection of neighborhoods in the alignment measure is then computed.

b) The main paper is showing results for only a single evaluation of **one** LLM (qwen3) being aligned to **one** vision and **one** audio encoder which can be interpreted in favor of the proposed hypothesis. While even in this single instance the alignment in the audio embedding is already quite weak, additional experiments in the appendix (fig. 19-25) hardly show any significant effects for other experimental setting using different LLMs  and datasets - especially in the audio case where often the *see* prompt reaches better alignment scores than the *hear* prompt. Even for the vision experiments it remains unclear if alignment improvements e.g. from 0.08 to 0.10 are significant.

c) the likely reason for the differences in the alignment of vision and audio models could be located in the LLM text training data: if this contains much more samples of textual descriptions of visual scenes than audio environments, it is much more likely that one could steer a model in this direction. Unfortunately, the authors are not investigating this point (which could be done by fine-tuning the LLMs with such descriptions and to measure the alignment effects).

In summary, I think that the paper raises an interesting question, however the current state of the paper lacks technical clarity and sufficient experimental conformation of the hypothesis.

**Questions:**

It would be interesting to also investigate multi-modal model (text-image or text-audio) to see if the additional modality during training effects the alignment abilities.

**Details Of Ethics Concerns:**

Mome

---

> ### Author Response · Authors · 2025-11-18
>
> Thank you for taking the time to write a thoughtful review! We are encouraged that you found the general question raised by our question interesting with practical impact. We kindly respond to your concerns and number them for convenience:
>
> # 1.
> > unclear technical description of the alignment measurement
>
> We apologize for the lack of clarity and have addressed this in the revision. This alignment measurement follows the definition introduced in Appendix A of Platonic Representation Hypothesis [1], which has been used in the methodology of recent works such as [2] and [3].
>
> # 2.
> > highly inconclusive experimental results which do NOT provide sufficient evidence for the proposed hypothesis
>
> We would like to clarify the scope of our paper. We are unsure what “proposed hypothesis” refers to here. In the introduction of the paper, we box our main finding in the quote. This is not a hypothesis or claim, but an observation we make in our paper by using the mutual k-nearest neighbors metric to quantify the degree to which two kernel representations align.
>
> We thank the reviewer for raising the concern about the experimental results. We acknowledge the limitation that alignment to the audio modality is less steerable, as clarified in the Limitations section. Our goal is to clarify when sensory steering works and when it does not, so we agree that the failure modes should be made clearer in the main text.
>
> To address generalizability, we extend the WiT alignment results for the Qwen3 family to the Qwen2.5-Instruct family (Figure 20), and we find that steering is reliable. We also add COCO image-text dataset experiments for Qwen3 on the vision-encoder side (Figure 21), and we note that on the vision side, steering is consistently stronger on WIT and COCO than on DCI. We now note this in Limitations.
>
> # 3.
> > Even for the vision experiments it remains unclear if alignment improvements e.g. from 0.08 to 0.10 are significant.
>
> Thank you for raising an important concern. We frame our contribution as showing that alignment can be steered at inference time, rather than as achieving large gains in absolute alignment. Generative representations already increase LLM–vision alignment relative to single-pass embeddings (e.g., 0.12 -> 0.14), and adding sensory prompting yields an additional shift (e.g., 0..14 -> 0.16).
>
> We also now put these numbers in the context of prior work. In the original PRH paper [1], Figure 3 shows mutual-kNN alignment between DINOv2 and language models ranging from 0.56B to 65B parameters; across this entire scale, alignment increases from roughly 0.10 to 0.16. In follow-up work [2], Figure 10 reports changes on the order of 1e-3 when varying the amount of structured reasoning data in pretraining. In this light, the inference-time shifts we observe for a single, fixed model across prompt conditions are even comparable to training-time interventions that have been studied.
>
> # 4.
> > the paper ignores the likely high influence of the LLM training data
>
> We agree that the training data influences the representations. In fact, that’s what we want: good visual and auditory representations only emerge if the models are trained on large, relevant datasets. Our hope is precisely that they have seen enough image/audio–related data to be capable of learning rich sensory structure. But being trained on relevant data does not automatically give you the specific behaviour we study here. It does not make it obvious an LLM’s internal representations can be steered toward a particular vision or audio encoder. The same was true in the original PRH paper: of course the structure is shaped by the corpus, but it is still a nontrivial question how that structure is organized as models become more performant.
>
> # 5.
> > It would be interesting to also investigate multi-modal model (text-image or text-audio) to see if the additional modality during training effects the alignment abilities.
>
> Thank you for raising this question. On the encoder side we already include models with multimodal supervision, such as CLIP and CLAP, as well as BEATs variants trained with audio–text supervision in Appendix C. We indeed observe higher baseline alignment with language, which is consistent with the fact that their objectives explicitly tie visual or auditory signals to text.
>
> —
>
> Thank you again for your helpful feedback! Do these answers address your concerns with the paper? If not, what further clarification or modifications could we make to improve your score?
>
> References:
> - [1] Huh, Minyoung, et al. "The platonic representation hypothesis." _Forty-first International Conference on Machine Learning_ (2024).
> - [2] Han, Junlin, et al. "Learning to see before seeing: demystifying LLM visual priors from language pre-training." _arXiv preprint arXiv:2509.26625_ (2025).
> - [3] Zhu, Tyler, et al. "Dynamic reflections: probing video representations with text alignment." _arXiv preprint arXiv:2511.02767_ (2025).

---

> > ### Comment · Reviewer_875U · 2025-11-19
> >
> > Thank you for your comprehensive response which clarifies some of my points:
> >
> > a) the  unclear technical description of the alignment measurement has been fixed in the revision to the extend that readers will be able to follow
> >
> > b) the newly added experiments now provide a better basis which give more support to generalize the results and the authors provide reference results which allow a better interpretation of the scales of the numerical results. However, the significance of the results is still weak in many cases, especially for the audio.
> >
> > c) The paper now discusses the influence of training data in more detail, which at least transparently shows the limitation ins this regard.
> >
> > Overall, I see a significant improvement of the paper. I'm still concerned that the overall effects shown are quite weak but will not oppose acceptance of the paper if the fellow reviewers give according scores.
> >
> > I'm raising my score to 4

---

> ### Author Response · Authors · 2025-11-19
>
> Thank you for your response. We appreciate your acknowledgement of the improvements we made in light of your concerns.
>
> We would like to clarify your main concern that the overall effects are weak. We agree that the absolute increases in alignment are small, roughly from 0.12 to 0.14 through generative representations and from 0.14 to 0.16 with the added sensory cue. However, this scale is comparable to the change reported in the original PRH paper, where increasing model size from 0.56B to 65B parameters raises alignment from about 0.10 to 0.16 (Figure 3 of PRH). As another point of reference, these two models (bloom-0.56B and llama-65B) differ dramatically in downstream performance, for example from roughly 30% to 70% on Hellaswag and from about 0% to 50% on GSM8K (Figure 4 of PRH). This suggests that modest shifts in alignment can correspond to substantial differences in behavior, and that observing such increases is nontrivial. In PRH and related work, changes of this size arise from large train-time interventions (more parameters, different data), so it is surprising to us that we can induce comparable alignment shifts purely at inference time by changing the prompt while keeping both the model and input captions fixed.
>
> Let us know if this helps address your concern!

---

> > ### Comment · Reviewer_875U · 2025-11-24
> >
> > psychologically, very low numerical changes like going fro, 0.14 to 0.16 are hard to interpret. As a reader, I would intuitively tend to discard from trying a method that shows improvements on that scale... However, if you can show clear downstream effects, this changes the picture. So, you could try to discus these effects more prominently and/or try to adapt the numerical scale. The latter heavily depends on the choice of k. Increasing k might give you easier to digest values. Did you perform an ablation on the choice of k?

---

> > > ### Author Response · Authors · 2025-11-24
> > >
> > > Thank you for the suggestion! You highlight concerns that extend beyond our specific setup and point to broader limitations of the methodology. Since this is a relatively recent line of work, these concerns would make meaningful directions for follow-ups. For our revision of the paper, we will contextualize the scale better by discussing the scale of the original PRH paper and corresponding downstream performance. In our work, we use $k=10$ and $n=1024$ in the mutual $k$-NN metric for the vision-language experiments as the original paper, which studies the effect of $k$ and other metrics in the appendix.

---

### Official Review · Reviewer_4qK6 · 2025-11-01

**Soundness:** 3
**Presentation:** 4
**Contribution:** 4
**Rating:** 8
**Confidence:** 4

**Summary:**

This submission demonstrates that “Sensory prompts” (prompts instructing an LLM to SEE or HEAR) bring the latent space of text-only LLMs closer to uni-modal specialist encoders in other domains (audio, vision) trained without any text supervision. This is shown by measuring representational similarity in a kernel space. Effectively, this paper is an empirical insight on the “platonic representation hypotheses” (Huh et al., 2024), where alignment of representations is not achieved by scale but explicitly steered into at test-time. Findings on main experiments include: LLM scale increases alignment (even under neutral cues) and steerability through cue separation; steering via sensory prompts increases alignment, whereas the opposite cue increases misalignment; longer generations (up to 256 tokens) increase alignment; the SEE cue even improves performance under "VQA without V"-kind of tasks

**Strengths:**

- The paper is exceptionally well written and presented
- This paper is an interesting twist on the “platonic representation hypotheses” (Huh et al., 2024), where alignment of representations is not achieved by scale but explicitly steered into at test-time. This topic is of significance to a wide audience.
- The experiments are well-designed (e.g., all encoders are trained without text supervision, test data seems not to be included in training) and carefully ablated (e.g., Sec. 3.3).
- The analysis is conducted over multiple axes of interest e.g., domains, scale, generation length.
- Beyond representational alignment, it is shown that SEE steering can even increase performance on certain tasks

**Weaknesses:**

- My biggest issue is that the paper presents cherry-picked (data, model) results, which give the illusion that all findings generalize beyond the shown test scenarios. That seems to be rather unclear. The appendix contains numerous examples where steering does not work as intended: e.g., in Figure 21 HEAR often performs better than SEE on DCI, and even the no cue is sometimes better. I understand why SEE doesn’t improve based on the explanation in the paper (DCI is already visually well described), but HEAR should not increase alignment. Another example is LLaMA 3.2 where audio is poorly steerable. Steering the opposite cue (e.g. HEAR for vision) often also increases alignment.
- The results in Sec. 3.6 are weakly supported by testing only one LLM.
- The submission is lacking a rationale why the average over all hidden states and tokens is a good representation. There is only some small support in the Appendix.
- Not all plots show error bars (Fig. 2/6)
- It’s not clear how the responses of Sec. 3.6 are scored. The model seems to be instructed to only answer yes/no (which would be trivial to score) yet the outputs below Figure 31 show lengthy outputs.



Minor presentation details:
- Axis labels overlap in Figure 15
- Label overlap in Figure 23 and 25
- Missing space after Figure 16 in L694
- “HEAR HEAR” in L1185

**Questions:**

- Can the authors reject test leakage, i.e., exclude that uni-modal samples from the respective test datasets were used during training of the models? I checked that DINO and BEATS do not contains the exact same datasets, but samples might still have been used.
- The effect of cue steering is sample dependent as evident from the error bars Is there any clear pattern in the samples that are more or less steerable?

---

> ### Author Response · Authors · 2025-11-18
>
> Thank you for taking the time to write a thoughtful review! We are encouraged that you found our paper well written, interesting, and significant. We also appreciate that you found our experiments well-designed over multiple axes of interest with downstream implications. We kindly respond to your concerns and number them for convenience:
>
> # 1.
> > My biggest issue is that the paper presents cherry-picked (data, model) results, which give the illusion that all findings generalize beyond the shown test scenarios.
>
> > The effect of cue steering is sample dependent as evident from the error bars Is there any clear pattern in the samples that are more or less steerable?
>
> We thank the reviewer for raising the concern about cherry-picked (data, model) combinations. Our goal is to clarify when sensory steering works and when it does not, so we agree that the failure modes should be made clearer in the main text.
>
> To address generalizability, we extend the WiT alignment results for the Qwen3 family to the Qwen2.5-Instruct family (Figure 20), and we find that steering is reliable. We also add COCO image-text dataset experiments for Qwen3 on the vision-encoder side (Figure 21), specifically to address your point about DCI. Across these settings, steering is consistently stronger on WIT and COCO than on DCI.
>
> We agree with your observation that DCI captions are already visually well described, and this likely explains why the SEE cue does not give much additional effect. DCI captions are long and dense with visual detail, while WIT and COCO captions are much shorter and have minimal sensory information. We posit there is more room for sensory prompting to change the representation when captions lack these details. (This also addresses your question about samples on which steerability is less reliable.) In contrast, prompting on an already visually saturated caption may not do much. For example, a HEAR-cued generation on a DCI caption still reads like a visual description:
>
> > …Starting with the grand temple. Golden columns—maybe they're tall and maybe there's some wind moving through them? But columns are usually solid, so maybe not. But maybe if there are any decorative elements or if the wind is causing some resonance. Vibrant roof tiles—maybe they're made of a material that clinks when wind moves them? Or maybe there's a gentle rustle if they're slightly loose. The central spire could catch the wind, maybe with some sort of wind chimes or just the sound of wind whooshing past…
>
> We also find that audio steering is less reliable overall, with alignment to audio encoders is lower than alignment to vision. We make this limitation more explicit in the main text.
>
> # 2.
> > The results in Sec. 3.6 are weakly supported by testing only one LLM.
>
> > It’s not clear how the responses of Sec. 3.6 are scored. The model seems to be instructed to only answer yes/no (which would be trivial to score) yet the outputs below Figure 31 show lengthy outputs.
>
> In the revision, we now evaluate three Qwen-family models (Qwen3-14B, Qwen2.5-Instruct-7B, Qwen2.5-Instruct-14B). Averaged across these models, the no-cue condition reaches $64.08 \pm 1.03\%$ accuracy and the SEE condition reaches $65.74 \pm 1.12\%$ (mean $\pm$ SE across models), an average gain of $1.7 \pm 0.4$ percentage points.
>
> We apologize that the scoring procedure was unclear. We now state explicitly that for scoring, we extract only the yes/no label from the output (ignoring the reasoning output), so evaluation reduces to a simple binary correctness measure.
>
> # 3.
> > Can the authors reject test leakage, i.e., exclude that uni-modal samples from the respective test datasets were used during training of the models?
>
> We agree that the training data influences the representations. In fact, that’s what we want: good visual and auditory representations only emerge if the models are trained on large, relevant datasets. Our hope is precisely that they have seen enough image/audio–related data to be capable of learning rich sensory structure. But being trained on relevant data does not automatically give you the specific behaviour we study here. It does not make it obvious an LLM’s internal representations can be steered toward a particular vision or audio encoder. The same was true in the original PRH paper: of course the structure is shaped by the corpus, but it is still a nontrivial question how that structure is organized as models become more performant.
>
> We agree that systematically varying the training data to study its effect on alignment and steerability would be interesting future work, but that is a different question from the prompt-level steering effects that are the focus of this paper.
>
> # 4.
> We have addressed the minor presentation details in the resubmitted manuscript.
>
> —
>
> Thank you again for your helpful feedback! Do these answers address your concerns with the paper? If not, what further clarification or modifications could we make to improve your score?

---

> > ### Comment · Reviewer_4qK6 · 2025-11-19
> >
> > Thanks for the rebuttal!
> >
> > > We also add COCO image-text dataset experiments for Qwen3 on the vision-encoder side (Figure 21), specifically to address your point about DCI.
> > Thanks for these results. The results on Qwen2.5 are very clear, especially at scale (btw, extra space in Line 1241), but on COCO they are fairly noisy again. Even for Qwen3 32B, I wouldn't be comfortable arguing for a strong steerability...
> >
> > > This also addresses your question about samples on which steerability is less reliable.
> > Okay, that is a simple case. I suppose there'll also be more descriptive captions from WiT/COCO, where steering still won't work. For instance, you could cluster them and see if there are any difference between steerable and non-steerable tokens.
> >
> > I am also curious (but this is maybe off-topic, so don't bother if you do not have time/compute): how similar are the vision representations of the four vision encoders to each other  (e.g. on WiT)  according to your similarity measure?

---

> ### Author Response · Authors · 2025-11-19
>
> Thank you for your response!
>
> > The results on Qwen2.5 are very clear, especially at scale (btw, extra space in Line 1241), but on COCO they are fairly noisy again. Even for Qwen3 32B, I wouldn't be comfortable arguing for a strong steerability...
>
> We agree. On COCO, steerability is much clearer for encoders with higher baseline alignment (DINOv2, CLIP) and noticeably weaker and noisier for ViT-MAE and ViT-MSN, with which Qwen3-32B aligns less well. In Figure 21, alignment to DINOv2/CLIP is more than twice that to ViT-MAE/MSN, which is consistent with the MAE behavior reported in the original PRH paper (Figure 3). This suggests that our conclusions about steerability should primarily be drawn from the better-aligned vision encoders, while treating results for weaker encoders and current open-source unimodal audio models more cautiously, since these audio models are still relatively less capable compared to their vision counterparts.
>
> > I suppose there'll also be more descriptive captions from WiT/COCO, where steering still won't work. For instance, you could cluster them and see if there are any difference between steerable and non-steerable tokens.
>
> Thank you for the suggestion! An a related experiment, we provide qualitative examples discriminating samples that benefit from the sensory cue vs samples that do not in Appendix B.
>
> > how similar are the vision representations of the four vision encoders to each other (e.g. on WiT) according to your similarity measure?
>
> Here, we report m-kNN between different the different vision encoders on WiT:
>
> |   | clip-vit-base-patch32 | dinov2-base | vit-mae-base | vit-msn-base |
> | --------------------- | --------------------- | ----------- | ------------ | ------------ |
> | clip-vit-base-patch32 | 1.0   | 0.476  | 0.273   | 0.259|
> | dinov2-base   | 0.476| 1.0 | 0.336   | 0.3433  |
> | vit-mae-base  | 0.273| 0.336  | 1.0  | 0.332   |
> | vit-msn-base  | 0.259 | 0.343  | 0.332   | 1.0  |
>
> Let us know if this helped answer your questions!

---

> > ### Comment · Reviewer_4qK6 · 2025-11-28
> >
> > Thanks for the response. The mean-kNN is indeed helpful to contextualize the alignment metric.
> > Overall, I really like this paper and want to retain my score. The findings are very interesting!

---

### Note · Program_Chairs · 2026-01-17
**Submission Desk Rejected by Program Chairs**

The following references in this submission do not refer to real documents and/or have major errors in bibliographic information:

 Qiushi Huang, Yuan Gong, Yu-An Chung, and James Glass. Masked autoencoders that listen. In Advances in Neural Information Processing Systems (NeurIPS), 2022.